

# Chemical and microbiological characterization of primary biological aerosol particles at the boreal forest

Jose Ruiz-Jimenez[1,5], Magdalena Okuljar[1,5], Outi-Maaria Sietiö[2,3,4], Giorgia Demaria[1], Thanaporn Liangsupree[1], Elisa Zagatti[1], Juho Aalto[4], Kari Hartonen[1,5], Jussi Heinonsalo[3], Jaana Bäck[4], Tuukka Petäjä[5] and Marja-Liisa Riekkola[1,5]

[1]Department of Chemistry, P.O. Box 55, FI-00014 University of Helsinki, Finland

[2] Department of Microbiology, P.O. Box 56, FI-00014 University of Helsinki, Finland

[3] Department of Forest Sciences, P.O. Box 27, FI-00014 University of Helsinki, Finland

[4]Institute for Atmospheric and Earth System Research/Forest Sciences, Faculty of Agriculture and Forestry, P.O. Box 64, FI-00014 University of Helsinki, Finland

[5]Institute for Atmospheric and Earth System Research, Faculty of Science, P.O. Box 64, FI-00014 University of Helsinki, Finland

*Correspondence to:* Marja-Liisa Riekkola (marja-liisa.riekkola@helsinki.fi)



**Abstract.** Primary biological aerosol particles (PBAPs) play an important role in the interaction between biosphere, atmosphere and climate, affecting cloud and precipitation formation processes. The contribution of pollen, plant fragments, spores, bacteria, algae and viruses to PBAPs is well known. In order to explore the complex interrelationships between airborne and particulate chemical traces (amino acids, saccharides), gene copy numbers, gas phase chemistry and the particle size distribution, 84 size-segregated aerosol samples from four particle size fractions (< 1.0, 1.0–2.5 µm, 2.5–10 µm and > 10 µm) were collected at SMEAR II station, Finland in autumn 2017. The gene copy numbers and size distribution of bacteria, *Pseudomonas* and fungi in PBAPs were determined by DNA extraction and amplification. In addition, free amino acids (19) and saccharides (8) were analyzed in aerosol samples by hydrophilic interaction liquid chromatography -mass spectrometry (HILIC-MS). Different machine learning (ML) approaches, such as cluster analysis, discriminant analysis, neural network and multiple linear regression (MLR) were used for the clarification of several aspects related to the PBAPs composition. Clear variations were observed for the composition of PBAPs as a function of the particle size. In most cases, the highest concentration values, gene copy numbers in the case of microbes, were observed for 2.5–10 µm particles followed by > 10 µm, 1–2.5 µm and < 1.0 µm. In addition, different variables related to the air and soil temperature, the UV radiation and the amount of water in the soil affected the composition of PBAPs. From the used ML approaches, especially MLR clearly improved the results achieved by classical statistical approaches such as Pearson correlation. In all the cases, the explained variance was over 91%. The great variability of the samples hindered the clarification of common patterns in the evaluation of the influence of microbes on the chemical composition of PBAPs. Finally, positive correlations were observed between the gas phase VOCs, such as acetone, toluene, methanol, 2-methyl-3-buten-2-ol, and the gene copy numbers of the microbes in PBAPs.



## 1 Introduction

Primary biological aerosol particles (PBAPs) can be defined as solid airborne particles directly emitted by the biosphere
into the atmosphere (Després et al., 2012). PBAPs constitute, expressed as mass concentration percentages, 30% of the
coarse particle fraction in urban and rural air (Fröhlich-Nowoisky et al., 2016), up to 65% at boreal forest (Manninen et
al., 2014) and even 80% in the case of the tropical forests (Elbert et al., 2007;Pöschl et al., 2010). PBAPs include both
dead and alive microorganisms (i.e. algae, archaea, bacteria, fungi and viruses), fragments or excretions from plants and
animals (i.e. plant debris and brochosomes) and dispersal units (i.e fungal spores and plant pollen) (Després et al.,
2012;Šantl-Temkiv et al., 2020).

Large aerosol particles are usually removed from the atmosphere, close to the emission area by dry deposition. However,
smaller ones have a relatively long residence time in air allowing their transport over long distances and interaction
processes (Reponen et al., 2001). The role of PBAPs in atmosphere can be very important affecting cloud and precipitation
formation processes by acting as cloud (CCN) and ice nuclei (IN) (Morris et al., 2011). However, their role is still poorly
understood because the sources and distribution of PBAPs in the atmosphere are not well quantified. Thus, to clarify their
atmospheric transport and ecosystem interactions, characterization and identification of chemical and microbial
constituents in PBAPs are needed (Spracklen and Heald, 2014).

Chemical tracers, such as free amino acids and saccharides, have been traditionally used for the determination of particles
of biological origin (Bauer et al., 2008;Helin et al., 2017). Free amino acids (AAs) are one of the most abundant
compounds in bioaerosols and they are also important markers for deposition and atmospheric transport (Barbaro et al.,
2011). Emission of AAs has been associated with the degradation of bacterial and biological materials (i.e. plants, pollens,
algae, fungi and bacterial spores, etc.), but they can be also related to volcanic emissions and compounds from combustion
(Dittmar et al., 2001;Ge et al., 2011). The exact role of amino acids in the atmosphere is barely known, but there might
be relation to the climate change and the atmospheric radiation balance (Chan et al., 2005). In addition, AAs can react
with other oxidants present in the atmosphere acting as pollutant scavenger or seeds for secondary aerosols (Haan et al.,
2009;Zhang and Anastasio, 2001).

Saccharides, frequently found in both urban and rural air (Yan et al., 2019), as primary saccharides (mono and
disaccharides), saccharide polyols (reduced sugars) and/or anhydrosaccharide derivatives (especially levoglucosan), are
often used as tracers for biomass burning. However, they are also closely related to fungal activity (Wan and Yu, 2007).
In this way, fungal saccharide emissions can correlate with several factors, such as temperature, carbon, nutrient and
oxygen availability (Pasanen et al., 1999). In addition, several studies have demonstrated the role of saccharides in the
formation of clouds, ice nuclei and regional climate change (Goldstein and Nobel, 1991;Goldstein and Nobel, 1994;Krog
et al., 1979).

Determination of chemical tracers in aerosol samples has clear advantages in qualitative analysis (Rathnayake et al.,
2017;Staton et al., 2015). However, this approach provides very little information about the microbiological
characterization of PBAPs (Gosselin et al., 2016;Zhu et al., 2015). Classical techniques such as cultivation and
microscopy, are widely used for the clarification of the different microbial groups present in the samples and can provide
information limited to viable and cultivable cells (Després et al., 2012;Manninen et al., 2014). Additional information,
about uncultivable, dead or fragments of plant and animal cells can be obtained using molecular genetic analysis



techniques, such as quantitative polymerase chain reaction (qPCR) or next-generation sequencing (NGS) (Després et al., 2012).

Viruses can be frequently found in the airborne attached to other suspended particles (Yang et al., 2011). In this way, PBAPs might be considered a potential route of viruses' infection and transmission (Pica and Bouvier, 2012). However, unlike other living organism such as bacteria, fungi and algae; viruses have not repair systems and their inactivation in

the atmosphere under the influence of different environmental factors should not be discarded (Després et al., 2012).

Bacteria, typical size range from 0.6 to 7.0 μm, can be found in the atmosphere as individual cells, attached to other particles or as an agglomerate (Lighthart, 1997). Bacterial air emission depends on many factors such as seasonality, meteorological factors, variability of bacterial sources and anthropogenic influence (Fang et al., 2018). Detailed aerosol–cloud models have shown that bacteria can alter the properties of clouds if present in sufficiently high number

concentrations (Phillips et al., 2009).

Fungi, fungal spores and their fragments are one of the most common components of PBAPs (Crawford et al., 2009). They have a common size range between 0.5 and 15 μm but also larger spores can be detected depending on environmental conditions, fungal species or age of the sporocarp (Huffman et al., 2010). Their role in the environment is of critical importance because many species can act as plant pathogens or trigger respiratory diseases and allergenic processes in

humans inducing considerable economic losses (Reinmuth-Selzle et al., 2017). In addition, global and regional models have been used to evaluate fungal spore emissions, transport and their impact on the hydrological cycle by acting as CCN and IN (Spracklen and Heald, 2014).

The expansion of computer methods allows researchers to use machine learning (ML) techniques, a branch of artificial intelligence, to clarify environmental issues (Liu et al., 2018). According to the datasets used for the development of

models, ML techniques can be classified into two broad categories: unsupervised ML, when data without labelling are used as input (i.e. cluster analysis (CA) and principal component analysis); and supervised MLs, using input (independent variables) and a target (dependent variable) attributes. In addition, supervised MLs can be used for the development of qualitative and quantitative models, the first using class labels and the latter continuous values as output variables (Smola and Vishwanathan, 2008). Discriminant analysis (DA), k-nearest neighbours, soft independent modelling by class analogy

and neural network (NN) are clear examples from algorithms used for the development of qualitative models. Multiple linear regression (MLR) and partial least squared regression are used for quantitative algorithms development (Rocha and Serrhini, 2018).

Even though the observations of concentrations and distribution of different PBAPs are accumulating, there is still lack of a comprehensive understanding of the processes behind the different observations and on detailed chemical

characterisation of the particles. Further, there are very few places where the airborne and particulate chemical and molecular genetic tracers, detailed gas phase chemistry and the particle size distribution can be simultaneously observed in field conditions. In this study, 84 size-segregated aerosol samples of four particle size fractions (< 1 μm, 1–2.5 μm, 2.5–10 μm and > 10 μm) were collected at the SMEAR II station (Station for Measuring forest Ecosystem-Atmosphere Relations, (Hari and Kulmala, 2005)) in Southern Finland in autumn 2017. The data from SMEAR II station allows

comparisons with >1200 simultaneously observed parameters from the forested ecosystem and the boundary layer, and thus provides a unique opportunity for comprehensive analysis of interrelationships. Chemical compounds (amino acids and saccharides), microbial species (bacteria, fungi and *Pseudomonas)* and total DNA concentrations were determined



by hydrophilic interaction liquid chromatography-mass spectrometry (HILIC-MS), and DNA extraction and amplification, respectively. Different statistical tools, including classical techniques and ML approaches such as CA, DA and NN were used to clarify the relationship between particle size, environmental and meteorological conditions and the composition of PBAPs. Pearson correlation and MLR were used for the elucidation of potential chemical signals from microbes in aerosol particles. Finally, the potential connections between gas phase VOCs and the microbiological composition of the aerosol particles, bacterial, fungal or *Pseudomonas* gene copy numbers were also evaluated using CA, DA and MLR.

## 2 Experimental section

### 2.1 Materials and reagents

Detailed information of materials and reagents is given in the electronic supplemental file (S1).

### 2.2 Instruments and apparatus

Aerosol samples were collected above the canopy (23 m) using a Dekati PM10 impactor (Dekati Ltd, Kangasala, Finland) which allows the simultaneous sampling of four particle size fractions (< 1.0, 1–2.5 μm, 2.5–10 μm and > 10 μm). 25 mm polycarbonate membranes from Whatman Nuclepore (Global Life Sciences Solutions, Pittsburgh, PA, USA) filters were used for the collection of the three largest particle size fractions. The smallest size fraction (< 1.0 μm) was collected on a 47 mm Teflon filter (Gelman Sciences LTD., Port Washington, NY, USA) with 2 μm pore size. To prevent particles from bouncing, membranes were smeared with diluted Apiezon L vacuum grease (Apiezon, Manchester, United Kingdom). A Branson 5510R-MT Ultrasonic Cleaner (Marshall Scientific, Hampton, NH, USA) was used for the extraction of saccharides and free amino acids from the filters.

An Agilent 1260 Infinity HPLC system (Agilent Technologies, Palo Alto, CA, USA) furnished with a SeQuant® ZIC®-cHILIC column (150 mm x 2.1 mm i.d., pore size 100 Å, 3 μm particle size) from MERCK coupled with an Agilent 6420 triple quadrupole mass spectrometer equipped with an electrospray ion source (Agilent Technologies), was used for the individual isolation and determination of amino acids and saccharides in a single HILIC-MS analysis. A KrudKatcherTM ULTRA HPLC In-Line Filter (0.5 mm) from Phenomenex (Phenomenex Inc., Torrance, CA, USA) was used to protect the column from potential particulate impurities.

Volatile organic compounds (VOCs) were measured by a high-sensitivity Proton-Transfer-Reaction Mass Spectrometer (PTR-MS, Ionicon Analytik GmbH, Innsbruck, Austria). The proton transfer reaction quadrupole mass spectrometer measured 13 different masses using a 2.0 s sampling time. Samples, were collected at 8.4 m above the ground level inside the canopy, continuous airflow was (43 L min$^{-1}$), were drawn down to the instrument using a heated 157 m line (14 mm i.d. PTFE tubing). From this line, a side flow of 0.1 L min$^{-1}$ was transferred to PTR-MS via a 4 m PTFE tube with 1.6 mm i.d. The instrumental background was determined every third hour by measuring VOC free air, produced with a zero air generator (Parker ChromGas, model 3501).



### 2.3 Sampling place and aerosol sampling

Samples and measurements were collected/conducted at SMEAR II station in Hyytiälä, southern Finland (61°51′N, 24°17′E, height above the canopy, ie 23 m, in autumn 2017 (4.09.2017-22.11.2017). SMEAR II is located in the middle of a forest that consists mostly of Scots pine (Pinus sylvestris L.) trees (Hari et al., 2013). In addition to Scots pine, there are some Norway spruce (Picea abies) and broadleaved trees such as European aspen (Populus tremula) and birch (Betula sp.). The forest is about 50-years old and the canopy height is currently ca. 18 m. SMEAR II is classified as a rural measurement station and there are no large pollution sources near the station. The nearest larger cities, Tampere (220 000 inhabitants) and Jyväskylä (140 000 inhabitants), are located about 60 km and 100 km from the measurement station. Otherwise, there are no large pollution sources nearby the station.

84 aerosol samples from 4 different particle size fractions (< 1.0 μm, 1–2.5 μm, 2.5–10 μm and > 10 μm; 21 from each) were collected, using a Dekati PM10 impactor. The sampling flow rate was on average 30 L min$^{-1}$ and the collection time was approximately 48 h (sampling volume 89–94 m$^3$). Additional information about the samples including sampling period and volume can be found in Table S1. After collection, filters were stored at –20°C inside a closed polystyrene Petri dish covered with aluminum foil. Before analysis, filters were cut into 2 pieces. One half was submitted to microbiological characterization of the samples by DNA extraction and amplification; and the second half was used for the simultaneous determination of amino acids and saccharides.

### 2.4 Chemical and microbiological characterization of the aerosol samples

The methods used for the chemical and microbiological characterization of the samples are based on those, described by Helin et al. in 2017 with some modifications. These changes are mainly related to the analytical procedure applied for the simultaneous determination of amino acids and saccharides.

### 2.4.1 Microbiological characterization

Total nucleic acids were extracted from the half of the filter selected for the microbiological evaluation using a commercial DNA extraction kit (PowerWater DNA Isolation Kit, MoBio Laboratories, USA). Briefly, DNA in the filters was extracted by 30 min incubation at 65 °C, using 1 mL of pre-warmed lysis buffer as extractant, and subsequent homogenization, vortexing horizontally for 2.5 min. The remaining steps were carried out according to the supplier´s protocol, and the DNA was eluted with 100μl of 10 mM Tris. The extracted DNA was further concentrated by precipitating it with 5μl of 3 M NaCl and 200 μl of 99% cold ethanol at –20°C for an hour before centrifugation at 10 000 × g for 5min. The liquid was decanted and the pellet was dried in ambient air before re-dissolving the DNA to 50μl of 10 mM Tris. The DNA concentration and purity was measured fluorometrically with Qubit 2.0 Fluorometer (Thermo Fisher Scientific, Waltham, MA, USA). The DNA samples were stored at −20 °C prior to qPCR.

The qPCR reactions were carried out with Bio-Rad CFX96 iCycler on 96-well white polypropylene plates (Bio-Rad, USA). The amounts of bacterial and fungal DNA in the samples were quantified with qPCR using target-specific primer pairs, Eub338F and Eub518R (Fierer et al., 2005), and FF390 and FR1 (Vainio and Hantula, 2000). Genus-specific primers, Eub338F (Fierer et al., 2005) and PseudoR (Purohit et al., 2003), were utilized to detect the bacteria belonging to the genus *Pseudomonas*. The bacterial and Pseudomonas genus specific reaction mixtures contained SsoAdvanced universal SYBR Green supermix (Bio-Rad, USA) at final concentration of 1x, 5 μL of template DNA, 250 nM of forward



and reverse primers. The reaction volume was adjusted to 20 μL with nuclease-free water. With fungal primers, the reaction mixture was otherwise the same, but the FF390 primer was in concentration of 250 nM and the FR1 primer in 200 nM. The qPCR reactions were conducted according to the manufacturer's protocol with combined annealing and extension (55 °C, 30 s for bacterial and Pseudomonas primers, and 60°C, 45 s for fungal primers). For quantification, fluorescence was measured during the elongation step. From each DNA sample and standard, the three technical replicates

were prepared, and from each mastermix three negative controls were analysed. The qPCR products were analysed in 1.5 % (w/v) agarose gel (BioTop) and visualized with 0.3 % (w/v) ethidium bromide (Sigma-Aldrich) under UV-light to ensure the correct amplicon length and the specificity of amplification. In the bacterial and Pseudomonas-specific qPCR reactions, standard curves were generated with DNA extracted from Pseudomonas fluorescens H-27 (Hambi culture collection, University of Helsinki), and for the fungal-specific qPCR, the DNA from the whole-genome-sequenced

Phlebia radiata FBCC43 (FBCC culture collection, University of Helsinki) was used (Kuuskeri et al., 2016). Results, expressed as gene copies, were normalized by the total amount of sampled air.

### 2.4.2 Chemical characterization

For determination of amino acids and saccharides, half of the filters were spiked with internal standard solutions and transferred to a test tubes for extraction assisted with ultrasounds at room temperature using 0.1% formic acid as extraction

solvent. Three extraction cycles (2 mL (15 min), 2 mL (10 min) and 1 mL (10 min)) were needed to ensure the complete extraction of the analytes. After removal of lipids and other non-polar interferences by liquid-liquid extraction (1 mL of hexane as extractant), the aqueous phase, approximatively of 5 mL, was concentrated to 1 mL with a gentle flow of nitrogen at 50 °C. A final centrifugation step was used to eliminate all possible non-soluble particles before analysis. The samples were stored at 4 °C and analyzed within 72 hours. Blank filters were processed simultaneously with the real

samples and used for the correction of the results.

The developed HILIC method allowed the simultaneous determination of amino acids and saccharides in the extracts. The column temperature was set to 50°C. The mobile phase A was acetonitrile with 0.1% formic acid, and the mobile phase B was MQ water with 0.1% formic acid. The separation of the target analytes was performed using following gradient program: 20% B for 15 min (0.4 mL min$^{-1}$), 20-80% B for 5 min (0.3 mL min$^{-1}$), followed by 80-20% B for 3

min (0.3 mL min$^{-1}$). The total analysis time was 23 min, with 12 min being required to re-establish the initial conditions. The injection volume was 3 μL. The entire effluent was fed to electrospray source for ionization (ESI; positive and negative mode for amino acids and sugars, respectively) and monitored by MS$^2$ detection in multiple reaction monitoring mode (MRM), with the exception of levoglucosan analyzed in selected ion monitoring mode. Ionization conditions and MRM parameters for the different compounds are found from Table S2. Results were normalized by the total amount of

sampled air.

### 2.5 Additional background data

Meteorological and environmental parameters are continuously measured at the SMEAR II station and the data are available from the AVAA portal (Junninen et al., 2009). From all the data available, more than 1200 simultaneously

observed parameters from the forested ecosystem and the boundary layer, 41 parameters, including aerosol parameters, concentration of atmospheric gases, meteorological and environmental data, were selected for further statistical analysis.



Detailed information about these variables is described in supplementary information (Table S4). Data from the portal (half-hourly averaged) were further averaged according to each sampling time-period by using arithmetic mean.

Volatile Organic Compounds (VOCs) were measured by a high-sensitivity Proton-Transfer-Reaction Mass Spectrometer.
Detailed information of the instrument used for the measurements, the measured masses, the potential relation between these mases and individual VOCs and the instrumental calibration are found from the electronic supplemental file S2.

### 2.6 Statistical analysis

A number of R, version 3.6.3, tools were used in this research for statistical analysis (Team, 2019). Skewness and Kurtosis
tests were used for the evaluation of the data distribution. Additional logarithmic transformation was needed to ensure normal data distribution of the input variables. Variables quantified in less than one fourth of the samples ($N < 5$) were not considered for further statistical analysis.

Pearson product moment correlations were evaluated between each pair of variables to measure the strength of their linear relationship. P-values were used to evaluate the statistical significance of the estimated correlations. P-values $\leq 0.05$
indicate statistically significant non-zero correlations at 95% confidence level.

Cluster analysis, a ML tool based on non-supervised pattern recognition approach, was used to group the samples into clusters according to the similarities in the meteorological and environmental data. The different variables used for the development of the model are listed in Table S4. In addition, samples were also grouped according to the gas phase emissions of VOCs. In all the cases, furthest neighbour method and squared Euclidean distance were utilized for model
development.

Linear discriminant analysis and probabilistic NN, based on Bayesian classifiers, ML algorithms were used to clarify the influence of the meteorological and environmental variables on the chemical and microbiological composition of the aerosol particles. The concentration of amino acids and saccharides (expressed as ng m$^{-3}$), the amount of bacteria, *Pseudomonas* and fungi (as gene copy numbers m$^{-3}$); and the total amount of DNA in the samples (as ng m$^{-3}$) were used
as input variables. These ML algorithms, DA and NN, were also used to elucidate the effects of the gas phase VOCs and the particle size on aerosol composition.

Multiple linear regression, a very simple ML approach, via backward stepwise selection to remove non-statistically significant variables, was used to evaluate the effect of the microbial species on the chemical composition of the aerosol particles. The same approach was selected for the clarification of potential gas phase VOCs connections to the
microbiological composition of PBAPs. The amount of bacteria, *Pseudomonas* and fungi, expressed as gene copy numbers m$^{-3}$, was used as dependent variable in all the cases. The concentration of amino acids and saccharides in the samples (expressed as ng m$^{-3}$) and the concentration of gas phase VOCs (expressed as ppbv) were used as independent variables. Different parameters were used for the evaluation of the MLR models. P-value will determine a statistically significant relationship between the variables at the 95.0% confidence level. The explained variance of the data was
provided by the R$^2$ values. Finally, the residuals of the model were evaluated using the standard error of the estimate, the mean absolute error and the Durbin-Watson test, which allows the elucidation of the prediction limit for new observations the average value and the potential autocorrelation between the residuals, respectively.



## 3. Results and Discussion

### 3.1 Analytical features of the methods used for chemical and microbiological analysis of the aerosol samples

The analytical approach used for the chemical characterization of the aerosol particles allowed the simultaneous determination of free amino acids and saccharides in a single HILIC-MS run. Typical chromatograms obtained for standard solutions and natural samples, the features of the method (regression coefficient, linear range, LOD and LOQ), extraction recoveries and the effect of the sample matrix in the results are found from electronic supplementary
information (Figure S1 and Tables S5–S7).

Good linearities were achieved for all the analytes under study. In addition, linear ranges of at least two orders of magnitude was achieved for most of the compounds under study. Tryptophan (Trp) provided the shortest linear range, one order of magnitude. On the other hand, glutamine (Gln), arginine (Arg) and fructose, provided a linear range of three order of magnitude. Detection limits, defined as the true amount of the analyte in the sample which will lead, with a
probability of 95%, to the conclusion that the concentration or amount of the analyte in sample is larger than in the blank, were calculated for each compound using the standard deviation of the estimate and the slope values (Shrivastava and Gupta, 2011). In the case of free amino acids, LODs ranged between 0.01 (Arg) and 0.04 (Histidine (His)) ng m$^{-3}$. Similar results were also achieved for saccharides whose LODs ranged between 0.01 (arabitol) and 0.05 (fructose) ng m$^{-3}$ (Table S5).

Recovery experiments were done using the approach proposed in Helin et al., in 2017. Extraction recoveries ranged from 73% (Tyrosine (Tyr)) to 134% (Serine (Ser)) for AAs, and from 82% (mannitol) to 125% (levoglucosan) for saccharides (Table S6).

A pool of samples containing different matrices (1.0–2.5 μm and > 10 μm particles), spiked at three different concentration levels (8, 77 and 111 ng of each analyte), were used to establish the accuracy of the method and the potential
matrix effects on the ionization. Average recoveries from 93.5% (Gln) to 116.4% (Asparagine (Asn)) and from 80.3% (fructose) to 99.6% (inositol) were obtained for amino acids and saccharides, respectively (Table S7).

In addition, the detection limit of the Qubit method for the determination of the total amount of DNA in the aerosol particles was 0.51 ng of DNA/filter.

### 3.2 Chemical and microbiological composition of the aerosol particles

Results achieved for the chemical and microbiological characterization of the aerosol particles collected from SMEAR II station in 2017 are shown in Figure 1 and Tables S8-S11. In general, the number and concentration of saccharides and amino acids as well as gene copy numbers increased with particle size until 10 μm.

A detailed study of the results found for the smallest, i.e. < 1.0 μm particles, reveals that bacteria and fungi were present
in almost all the samples. However, *Pseudomonas* was found just in 50% of them. In addition, the total DNA concentration was under the LOQ for 66% of the samples. The average gene copy numbers of bacteria were higher than fungi. The chemical composition of < 1.0 μm particles was quite simple in comparison with bigger particle sizes. Just five free amino

**Figure 1.** Chemical and microbiological composition of the analyzed filters. Start sampling day was represented in x-axis. Bac, bacteria; Pse, *Pseudomonas*; and Fun, fungi.





acids, alanine (Ala), glutamic acid (Glu), glycine (Gly), valine (Val) and proline (Pro) were found at least in 70% of the samples. From these, Gly and Pro were at the highest and lowest concentration, respectively. The relevance of saccharides was very limited in this fraction, only arabitol and mannitol were found in approximately half of the samples. These results are in good agreement with those found for remote source aerosol particles (Scalabrin et al., 2012).

In the case of mid-size range, i.e. 1.0–2.5 µm particles, the total concentration of DNA was over the LOQ at least in 50% of the samples. In addition, compared to < 1.0 µm particles the chemical complexity of the samples clearly increased. Arabitol, mannitol and trehalose were found at least in 70% of the samples. Mannitol and threalose were the saccharides present at the highest concentration. A detailed examination of the results for free amino acids showed the presence of two different groups as a function of the detection frequency in the samples under study. Gln and Glu, Arg, Ala, Pro and phenylalanine (Phe) were detected at least in 70% of the samples. However, His was found only in 50% of them. Finally, Gln and Phe were the amino acids found with the highest and lowest concentration, respectively.

Microbiological results gave clear differences between the large particles, i.e. 2.5–10.0 µm particles and the other size classes. Exceptionally high gene copy numbers were determined for fungi and total DNA in all the samples. The results provided by the chemical analysis of the samples demonstrated a clear increase in the number of saccharides (7) and free amino acids (11) found at least in 70% of the samples, compared with the smaller size classes. Also in these particles mannitol and threalose were the saccharides detected with the highest concentration. In addition, Arg and Trp were the amino acids with the highest and the lowest concentration, respectively. These results, including those achieved for 1.0–2.5 µm particles, are in good agreement with those reported previously for aerosol particles collected from SMEAR II station (Helin et al., 2017).

Finally, the microbiological analysis of largest particle size class, i.e. > 10 µm particles showed that bacteria, fungi and total DNA were above LOQ in all the samples under study. As observed for smaller sizes, the gene copy numbers of bacteria were higher than those of fungi also in this size class. In addition, *Pseudomonas* were detected just in 40% of the samples. Focusing on the chemical analysis, the number of saccharides detected at least in 70% of the samples decreased to 4 in comparison with the 2.5–10.0 µm particles where 7 saccharides were detected. This can be explained by considering a clear decrease of the fungi gene copy numbers in $\geq 10$ µm particles and the role of saccharides as typical fungal markers (Bauer et al., 2008). However, the number of amino acids present at least in 70 % of the samples remained the same (11) as in smaller particles. In this case, Trp and leucine (Leu) were the amino acids with the highest and the lowest concentration, respectively. Leu is one of the most abundant AAs in plants, which could indicate its participation in > 10 µm particles though plant debris and pollen grains emissions to the air (Mashayekhy Rad et al., 2019;Nicolson and Human, 2013).

### 3.3 Evaluation of the relationship between the particle size and the composition of PBAPs

The relationship between different particle sizes for the microbial groups and chemical compounds detected in PBAPs was evaluated using two different statistical approaches. The first one, based on classical correlation (Pearson), was used to find statistically significant relations between the particle sizes for the individual species (chemical and microbial). The concentration of the individual compounds, number of gene copies for the microbes, determined for the different particle





sizes were used as variables. The second method allowed the clarification of potential variations on the composition of different size PBAPs based on the use of microbiological and chemical profiles. This approach required the use of more

complex statistical algorithms (ML approaches) such as, DA and NN. In this case, 84 samples were divided into training and validation sets, containing 68 (80%) and 16 (20%) samples, respectively. These sets were used for development of the models and subsequent unbiased evaluation, respectively.

The evaluation of the potential correlations between different particle sizes, using as variables the individual chemicals and microbes, can be useful to identify the presence of a common emission source or growing mechanism. However, a

surprisingly low number of correlations were found between the different particle sizes for the individual compounds and microbial groups under study (Figure S2). These results can be caused by the large number of potential PBAP sources present at the sampling site and their different contribution to the atmospheric aerosols. None of the evaluated variables, chemical or microbial, provided simultaneous correlations between all the particles sizes under study. In addition, the highest number of correlations between particle sizes were achieved for fungi and Gln. In both cases, it was possible to

find correlations between the samples with a particle size shorter than 10 μm. Additional correlations for different compounds were observed between 1.0–2.5 μm and 2.5–10.0 μm particles (Figure S2).

Discriminant analysis was able to provide correct classification of 61 (89.7%) and 14 (87.5%) of the samples included in the training and validation sets, respectively. A detailed study of the influence of the variables on the model indicated three different trends (Figure 2). The first, found for *Pseudomonas* gene copy numbers, inositol, His and Pro, showed an

increase of the concentration with the particle size. The second, observed for levoglucosan and Gly revealed the highest concentrations for the smallest particle size (< 1 μm) and then a drop to the lowest concentration values for 1.0–2.5 μm particles followed by a progressive increase of the concentration with the particle size. The last one, found for most of the compounds and the microbial groups, gave the highest concentration values for 2.5–10 μm particles followed by > 10 μm and 1–2.5 μm. The concentrations of the different chemicals and gene copy numbers of microbial groups in the latter,

1–2.5 μm, were higher or at least similar to that achieved in the smallest particle size under study (< 1 μm) with some exceptions.

The use of a more complex ML algorithm such as NN for the classification of the samples allowed the correct classification of 52 (76.5%) and 12 (75.0%) of the samples included in the training and validation sets, respectively. The relative limited number of samples used for the development of the different models have a clear influence on their

classification performance.





**Figure 2.** Comparison of the concentration profiles obtained for the different chemical and microbiological species and

total DNA found in the filters as a function of the particle size. ■, < 1.0 μm particles; ■, 1.0–2.5 μm particles; ■, 2.5–10.0

μm particles; and ■, 10 > μm particles. Bac, bacteria; Pse, *Pseudomonas*; and Fun, fungi.





**3.4 Influence of the concentration of atmospheric gases, aerosol, meteorological and environmental parameters on the microbiological and chemical composition of the aerosol particles**

The influence of atmospheric gases concentration, aerosol, meteorological and environmental parameters on the atmospheric aerosol particles composition was evaluated using a two-step approach.

First, samples were classified into different sampling periods according to the differences observed for the concentration of atmospheric gas, aerosol, meteorological and environmental parameters. From all the parameters, simultaneously monitored at SMEAR II station (>1200) for the forested ecosystem and the boundary layer, 41 (Table S4) were selected

as CA data input for the visualization of seasonal differences between samples. Farthest neighbour approach and squared Euclidean distance were applied for clustering. The final number of clusters, sample groups, was optimized according to the distribution provided by CA, avoiding the presence of clusters containing a very limited number of samples, which would hinder the development of additional statistical analysis (Table S12).

Two different sampling periods (clusters) were found for the campaign under study. The first period (group 1) contains

48 samples collected from 04 September to 13 October, 2017. The second period (group 2) consists of 36 samples collected from 23 October to 22 November, 2017. Differences between periods, based on the different variables used in this study, can be found in Figure S3. The main differences were observed for snow (M21) and water precipitations (M22), the temperature of both air (M14) and soil, the later at different layers (M33–M36), the solar radiation (M23–M25) and other parameters such as evapotranspiration (M03), soil heat flux (M41) and gross primary production derived

from net ecosystem exchange (M05). For these variables, the highest values were found for group 1 samples, with the exception of the snow precipitation.

Once differences between the different sampling periods were established, ML tools such as DA and Bayesian NN were used to clarify the influence of the concentration of atmospheric gases, aerosol, meteorological and environmental parameters on the chemical and microbiological composition of PBAPs. Sample groups obtained from the previous step

were used for the classification of the aerosol particles using their chemical and microbiological composition as input data. As stated in the previous section, samples (84) were divided into training and validation sets containing 68 (80.0%) and 16 (20.0%) samples, respectively.

Discriminant analysis provided the correct classification of 60 (88.2%) and 13 (81.2%) of the samples included in the training and validation sets, respectively. In addition, 5 (45.5%) of the samples incorrectly classified were found in < 1.0

μm particles. This could be explained by the limited information achieved for the chemical and microbiological composition of these samples (Figure 1). As can be seen from Figure 3, a clear trend was found for the average concentrations calculated for all the particle sizes under study. The gene copy numbers of bacteria and fungi, and concentration of chemical species were higher in the PBAPs collected during Period 1 compared to Period 2. The same trend was observed for the individual particle sizes. In addition, these results were supported by the use of Bayesian NN

for the classification of the samples allowing the correct classification of 49 (72.1%) and 10 (62.5%) of the samples included in the training and validation sets, respectively. From all the samples incorrectly classified, 11 (43.8%) of the samples belongs to < 1.0 μm particles. Once again, the classification performance of more complex algorithms such as NN is clearly affected by the limited number of samples analyzed. The use of a larger dataset would improve the reliability of these results.




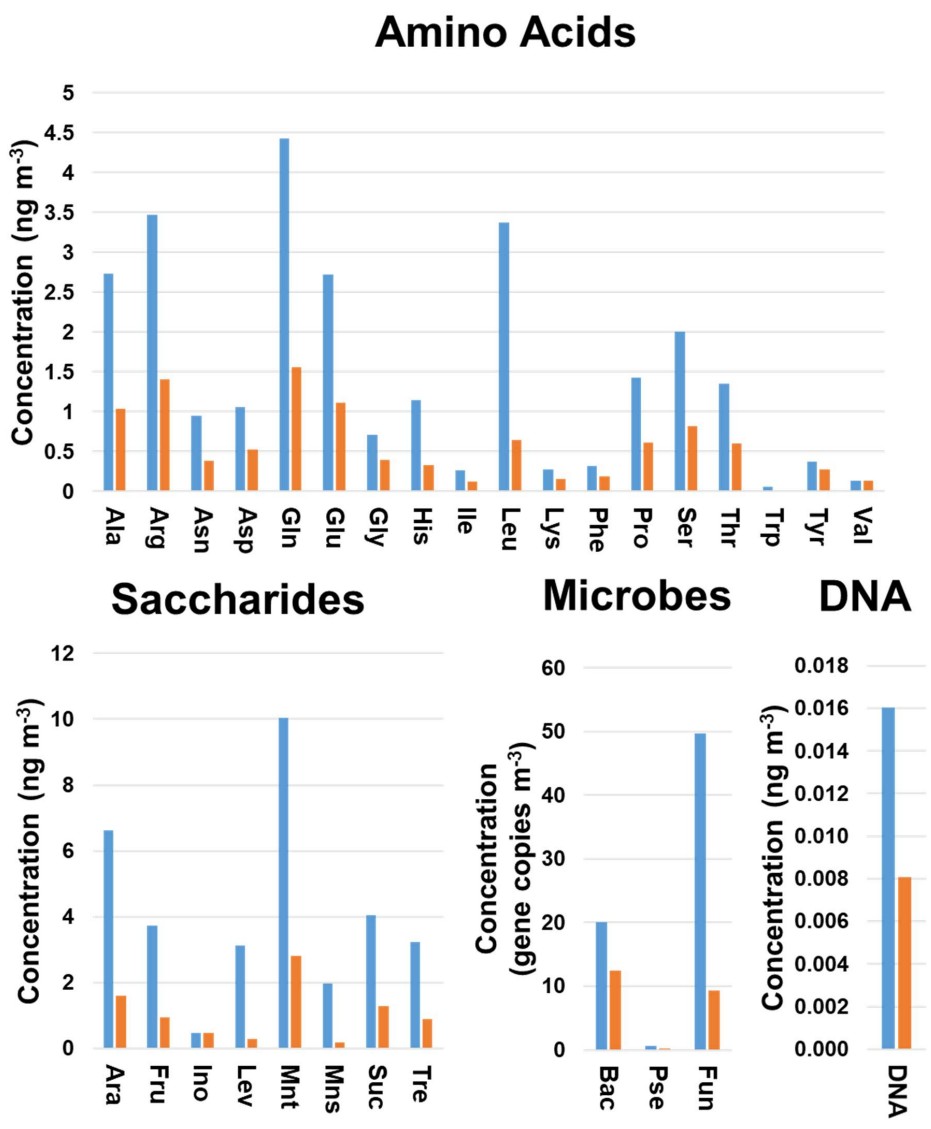

**Figure 3.** Differences in the chemical and microbiological composition of the PBAPs according to the sampling period. Average concentrations were calculated for all filter sizes analyzed. ■ Sampling period 1 (04.09.2017 to 13.10.2017) and ■ Sampling period 2 (23.10.2017 to 22.11.2017). Bac, bacteria; Pse, *Pseudomonas*; and Fun, fungi.


**3.5 Potential elucidation of chemical signals from microbes in aerosol particles**

It is well known that different chemicals, such as saccharides and amino acids are produced and in some cases emitted to the atmosphere by biological sources like bacteria and fungi via metabolic activities (Bauer et al., 2008). These chemical compounds might act as a clear indicator of the microbes' existence in PBAPs or at least of the presence of a common emission source. A simple way to elucidate potential chemical signals from microbes in PBAPs is the evaluation of the correlation between the concentration of the chemical species and the number of gene copies of multiple microbes in the

aerosol particles. However, this approach has clear limitations diverted from the presence of multi-emission sources in the sampling place for the chemicals under study. In this way, the unique and exceptional characteristics of the SMEAR II station, a remote measurement station with no large pollution sources nearby, ensures the quality of the samples used in the development of the different studies.

Two different approaches, based on statistical analysis of the results, were selected for the elucidation of the potential

chemical signals from microbes in the aerosol particles. The first one, based on Pearson correlation, uses the individual microbial groups (bacteria, *Pseudomonas* and fungi) and chemical compounds as dependent and independent variables, respectively. The second one, based on MLR, uses the individual microbial groups as dependent variables and the chemical profiles, including AAs and saccharides, as independent variables.

The irregular occurrence of chemical compounds and microbes present in the aerosol particles of different sizes (Figure

1) hinders the development of general statistical models containing all the samples under study. Individual models for the different particle sizes were evaluated to minimize these problems. In addition, the limited number of samples used for the development of the different models (<20) could affect the reliability of the results, including their predictive capacity. However, these MLR models are especially useful for the identification of the potential microbiological origin of the different AAs and saccharides present in the sample.

The results achieved for the Pearson correlations between chemical compounds and microbial groups (P-value < 0.05) can be seen in Figure S4. In the case of MLR, backward stepwise selection approach was used to remove non-statistically significant variables. This approach is especially relevant considering the potential limitations of MLR. As stated before, the concentration of the different amino acids (19) and sugars (8) present in the aerosol particles were used as independent variables. When the number of independent variables was higher than the number of samples, the variables were split

into two shorter batches, each of them containing 50% of the variables. Preliminary MLR models were developed to identify statistically significant variables, which were subsequently selected for the development of definitive equations. Detailed information of the different developed MLR models can be found in Figure 4 and Table S13.

The evaluation of Pearson correlations for < 1.0 μm particles (Figure S4) gives positive correlations between all the microbial groups and Pro. The simultaneous use of different AAs and saccharides for the development of the MLR

equations clearly improve the results in comparison with Pearson correlation (Table S13). The evaluation of the statistically significant MLR coefficients (Figure 4) indicates clear correlations between the gene copy numbers of the microbial groups, the concentration of Pro (not for *Pseudomonas*) and different saccharides such as arabitol and fructose. As stated before, saccharides are produced by biological sources like bacteria and fungi via metabolic activities (Bauer et al., 2008). Specifically, arabitol is well known as fungal spore tracer (Jia and Fraser, 2011;Yang et al., 2012). In addition,

Pro is a widely recognized degradation product of the organic matter by bacteria and fungi (Li, 2019).

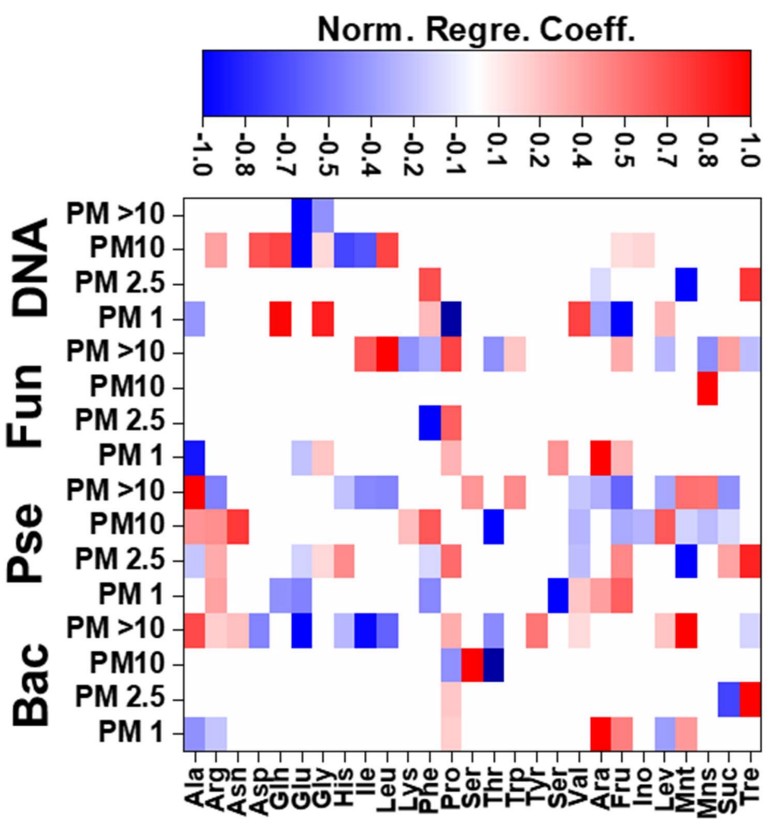

**Figure 4.** Normalized regression coefficient obtained for the MLR models developed for the potential elucidation of chemical signals from microbes in the PBAPs. Maximum normalization was applied in all the cases. Bac, bacteria; Pse, *Pseudomonas*; and Fun, fungi.


In the case of 1.0–2.5 μm particles, it should be emphasized that Pearson correlations indicated a clear negative correlation between the total DNA concentration and His (Figure S4). This could be explained by the role of the AA on the oxidant-induced DNA damage of the organisms (Cantoni et al., 1992). Additional positive correlations were observed between

microbial groups and Glu, Gln and mannitol. The first is a well-known fermentation by product of bacteria and fungi (Kinoshita et al., 2005). Mannitol, synthesized by plants, is released into bacteria and fungi during the degradation of the organic matter (Upadhyay et al., 2015). In addition, Gln, a metabolite of central importance in bacterial and fungal physiology, is the base of a large variety of nitrogen-containing compounds and close related with bacterial and fungal ammonia assimilation (van Heeswijk et al., 2013). It should also be noted, that MLR clearly improves the correlation

results in comparison with Pearson correlation (Table S13). The detailed evaluation of the regression coefficients (Figure 4) indicated that trehalose correlates with the gene copy numbers of bacteria and *Pseudomonas*. Finally, Phe correlates with the gene copy numbers of *Pseudomonas* and fungi. The presence of trehalose has been reported for a wide variety of microorganisms, including bacteria, yeast, fungi and insects (Elbein et al., 2003). Finally, microorganisms utilize Phe for the production of more complex compounds as cinnamic acid (Hyun et al., 2011).



The correlation between sucrose and the microbial groups could be of special interest. This saccharide has a key role in the symbiotic association between bacteria and fungi with plants (Vargas et al., 2009). The evaluation of the results, achieved from Pearson correlation studies in the case of 2.5–10.0 µm particles (Figure S4), gave a positive and weak correlation between bacteria, fungi, Gln and sucrose. This correlation was previously observed for 1.0–2.5 µm particles. The use of more complex statistical approaches, such as MLR, could explain higher number of data variability in

comparison with Pearson correlation (Table S13). The evaluation of the regression coefficients with a statistically significant influence on the MLR models (Figure 4) showed a clear correlation between mannose and the gene copy numbers of fungi. The role of mannose in fungi as cell wall component is very important, providing integrity and microbiological viability (Meyer-Wentrup et al., 2007).

Finally, detailed evaluation of the results performed for the aerosol particles with a diameter over 10 µm gave correlations

between bacteria and saccharides, such as mannose and trehalose (Figure S4). Mannose is used by bacteria as lectin like substances to bind to cells and perhaps to find its way also to aerosol particles (Sharon et al., 1981). Additional correlation was found between bacteria and some important AAs in the Krebs cycle. *Pseudomonas* had a clear correlation with Gly. The simplest AA, Gly, is used as a metabolic product in some bacteria. Specifically, Gly is used as carbon source in the biosynthesis of complex structures in the case of *Pseudomonas* (Lundgren et al., 2013). The explained variance of the

data was clearly improved by the use of MLR models, as can be seen in Table S13. The detailed evaluation of the regression coefficients (Figure 4) showed a clear positive correlation between gene copy numbers of bacteria and *Pseudomonas* in the samples and the concentrations of mannitol and Ala. The latter has important role in bacteria, especially in *Pseudomonas*, as a constituent of pantothenic acid (Boulette et al., 2009). In addition, it was possible to find significant correlations coefficients between fungi and three AAs such as Ile, Leu and Pro. These AAs, essential for

humans, can be easily synthetized by fungi (Jastrzębowska and Gabriel, 2015).

### 3.6 Elucidation of potential connections between gas phase VOCs and the microbiological composition of the aerosol particles

Soil microorganisms, bacteria and fungi, are able to produce large quantities of highly diverse VOCs diverted from

organic matter mineralization and accumulation processes. Some of these VOCs are emitted to the atmosphere. Once in the atmosphere they can be intake by other organisms, released in the underground habitat or participate in the formation and growing of secondary aerosol particles. In this way, the evaluation of the potential connections between gas phase VOCs and the microbiological composition of the aerosol particles might provide information about their role as potential emission sources or at least common emission conditions for both VOCs and microbes.

Preliminary studies were carried out following the two-step methodology previously described (section 3.4). First, the gas phase concentrations of 13 VOCs (Figure S5) were used for the development of CA models. These models allowed the visualization of seasonal differences between samples. The samples were collected during two different time periods (clusters). The first 32 samples were collected from September 4 to September 15, 2017 and from September 25 to October 6, 2017 (Period 1). Then 52 samples were collected from September 18 to September 22, 2017 and from October 9 to

November 22, 2017 (Period 2). In all the cases, the highest concentrations of VOCs in gas phase were found during Period 1 (Figure S6a). These periods were used for the development of DA models, which allowed the classification of the aerosol particles using their chemical and microbiological composition as input data. Discriminant analysis was able to





provide correct classification of 72 samples, 85.7% of the total. Period 1 samples provided the highest values for all the chemical and microbial groups with the exception of fructose, confirming a clear connection between the gas phase

concentrations of VOCs and the presence of chemical and microbial groups in the aerosol particles (Figure S6b). These results were quite similar, only small variations were seen, compared to those found using the concentration of atmospheric gases, aerosol, meteorological and environmental parameters as variables. In this way, the connection between these variables, the gas phase VOCs and the chemical and microbiological composition of the aerosol particles is clear.

The second approach based on the statistical tools was used to evaluate the correlations between the concentration of gas phase VOCs and the gene copy numbers of bacteria and fungi in the aerosol particles by Pearson correlation and MLR algorithms. The amount of the different microbes, expressed as number of genes copies m$^{-3}$, was used as dependent variables. The gas phase concentration of 13 VOCs was used as independent variables. The limited number of independent variables used in this study, smaller than the number of samples in all the cases, allowed the use of ordinary least squares

as fitting algorithm, instead of the backward stepwise algorithm used in the previous section. The limitations diverted from the small number of samples used for the development of the different statistical models, discussed in the previous section, were considered in the present study.

The results achieved form Pearson correlations can be found in Figure S7. The detailed evaluation of these results for < 1.0 µm particles (Figure S7) gave positive correlations between *Pseudomonas* and seven of the VOCs analyzed. However,

no correlation with the gas phase VOCs was observed in the case of other bacteria and fungi. In most of the cases, exception of methacrolein, these compounds were reported to be produced and emitted to the atmosphere by *Pseudomonas* (Effmert et al., 2012). As stated before, the use of relatively complex ML approaches, such as MLR, clearly improved the results. Successful models were achieved for all the microbial groups under study. Detailed information of the MLR models can be found in Table S14. The evaluation of the statistically significant MLR coefficients (Figure 5) indicated

that air concentrations of acetone, toluene and isoprene correlate with the variations in the gene copy numbers of the different microorganisms. All these compounds can be produced and emitted by a large variety of bacterial and fungal species (Effmert et al., 2012). However, additional emission sources might not be discarded.

Multiple correlations (Figure S7) were found between bacteria-fungi and different VOCs (6) present in the gas phase in the case of 1.0–2.5 µm particles. As can be seen in Table S14, the use of a combination of variables in the MLR algorithm

clearly improved these results in comparison with Pearson correlation. The detailed evaluation of the regression coefficients for MLR (Figure 5) revealed a clear correlation between the concentrations of isoprene and benzene and the variations in the gene copy numbers of the different microorganisms. It should be emphasized the correlations observed for the later. It is widely believed that benzene is just emitted by anthropogenic sources. However, several bacterial and fungal species such as *Bacillus simplex* (Gu et al., 2007)*, Acremonium obclavatum* or *Aspergillus versicolor* (Ezeonu et

al., 1994) can also produce and emit this compound to the atmosphere. In addition, ethanol, acetic acid and toluene correlated with the variation observed just in bacteria and fungi.

It was not possible to find any correlation between the concentration of the VOCs in the gas phase and the microbial groups detected in 2.5–10.0 µm particles. However, successful models were achieved in the case MLR as can be observed in Table S14. The analysis of the regression coefficients (Figure 5) indicated that gas phase concentration of ethanol,

acetone, 2-methyl-3-buten-2-ol and acetic acid correlate with the variations in the gene copy numbers of bacteria and fungi. These results are in good agreement with those achieved for 1.0–2.5 µm particles. However, the opposite trend was



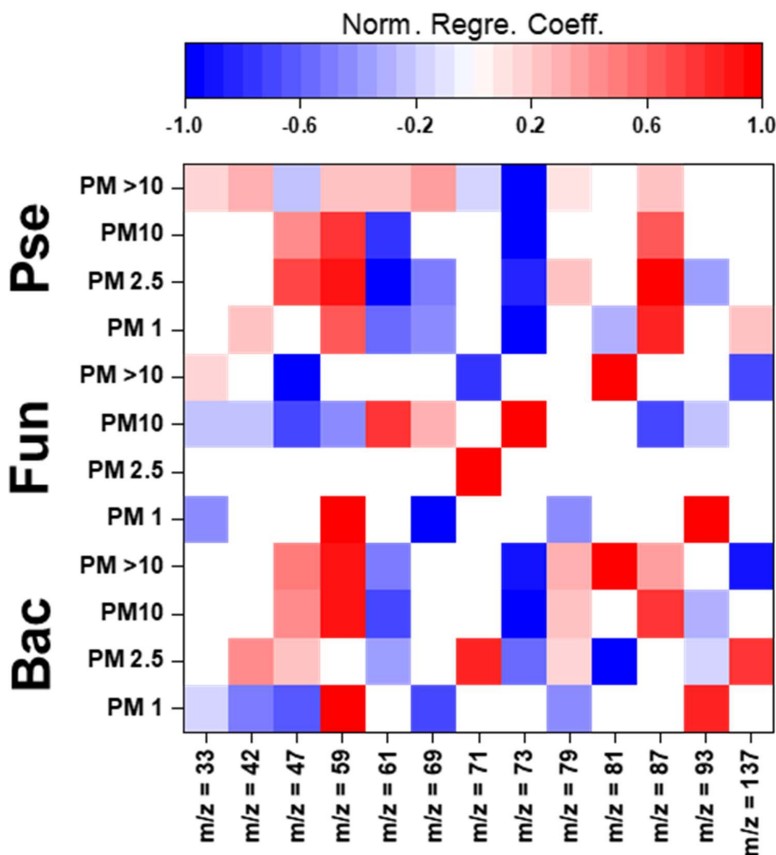

**Figure 5.** Normalized regression coefficient obtained for the MLR models developed for the elucidation of potential connections between gas phase VOCs and the microbiological composition of the PBAPs. Maximum normalization was applied in all the cases. Bac, bacteria; Pse, *Pseudomonas*; and Fun, fungi.


observed in the case of *Pseudomonas*. A potential explanation might be the wide variability of *Pseudomonas* present in the samples (Fröhlich-Nowoisky et al., 2016).

Finally, weak positive correlations (Figure S7) were found between bacteria, fungi and several VOCs such as acetone and

isoprene in the case of PBAPs with a diameter over 10 µm. It should be, however, emphasized that the sign of the correlation in the case of isoprene was opposite to that observed for < 1.0 µm and 1.0–2.5 µm particles. The use of MLR algorithms improved the results obtained from Pearson correlations (Table S5). The evaluation of the statistically significant regression coefficients (Figure 5) revealed some interesting results. The concentration of different VOCs such as acetone, 2-methyl-3-buten-2-ol, benzene and 2-butanone correlates with bacteria and fungi gene copy number

variations. In addition, terpenes, including monoterpene fragments, correlates with the variations in the gene copy numbers of bacteria and *Pseudomonas.* The role of plants in the production and emission of monoterpenes to the atmosphere is well known,. However, some species of bacteria, including *Pseudomonas*, can synthesize and emit these



compounds to the air (Effmert et al., 2012). In this way, it is not possible to discard the contribution of bacteria to the
monoterpene concentration or the role of environmental and meteorological parameters in the common emission of
terpenes and bacteria to the atmosphere.

## 4. Conclusions

Chemical compounds and microbial species were determined in aerosol particles collected from SMEAR II station in
September-November 2017. The use of these unique observations, integrated to the abundant chemical, meteorological
and environmental information as input data for the development of different statistical models, including classical
techniques and ML approaches, allowed the elucidation of several aspects related to the composition of PBAPs.

The use of different approaches, based on ML as DA or Bayesian NN, confirmed a clear relationship between the
composition of the aerosol particles and the sizes. Different trends were observed for the distribution of the chemical
compounds and microbial species between the samples. These results were surprising considering the relative low number
of correlation observed using traditional Pearson correlation. It should be also emphasized that the limited number of
samples analyzed can affect the performance of Bayesian NN decreasing the prediction capacity.

In addition, ML approaches allowed the clarification of the effect of concentration of atmospheric gases, aerosol,
meteorological and environmental parameters on the aerosol particles composition. The different conditions observed for
the two sampling periods stablished by CA were clearly connected to the variations observed for the chemical and
microbial composition of the aerosol particles. High concentrations were observed for the different chemical compounds
and microbes (gene copy numbers) in the samples collected during the period 1. The quality and quantity of the data used
had a clear influence on the model performance affecting specially those belonged to < 1.0 µm particles.

The elucidation of chemical signals from microbes in PBAPs was possible due to the exceptional characteristics of the
SMEAR II station, a rural measurement station with no large pollution sources nearby, which minimize the presence of
interferences in the study. These signals might be related with the presence of the chemical compounds in a microbial
species or at least a common emission source. The complexity of the system due to the great variety of the microbial
species present in the PBAPs and the potential emission sources hinder the clarification of common trends for all the
particle sizes under study. Multiple connections between the microbial and the chemical composition of the aerosol
particles were observed for the different sizes using classical and more advance statistical approaches. These connections
are mainly associated to metabolic processes but in some cases they are also related to protection or interaction
mechanisms.

Finally, the elucidation of potential connections between gas phase VOCs and the microbiological composition of the
aerosol particles confirmed a clear relationship between the VOCs in the gas phase and the presence of chemicals and
microbes in the aerosol particles. This might confirm the presence of common emission sources or at least emission
conditions. The results achieved are promising, even if the number of samples used for the model development was very
limited. Additional studies are needed to provide more reliable statistical models that could provide proxy concentration
of aerosol particles in the boreal environment that relies on the more standard observations performed continuously on
the site.




**Data Availability**

Data used in this work are available from the authors upon request (marja-liisa.riekkola@helsinki.fi).

**Author contribution**

Author contributions. JR-J, OMS, JH, JB, KH, TP and M-LR designed the experiments. JR-J, MO, OMS, GD, EZ, JA and TL carried them out. JR-J and OMS performed the statistical analysis. JR-J, MO, OMS KH, JB, TP and M-LR prepared the manuscript with contributions from other co-authors.

**Competing interests**

The authors declare that they have no conflict of interest.

**Acknowledgements**

Financial support was provided by Academy of Finland Centre of Excellence program (project no. 307331). We acknowledge financial support via Academy of Finland NANOBIOMASS (decision number 307537), Biogeochemical 620 and biophysical feedbacks from forest harvesting to climate change (decision number 324259), Molecular understanding on the aerosol formation in the high Arctic (decision number 333397), Belmont Forum project "Community Resilience to Boreal Environmental change: Assessing Risks from fire and disease" (ACRoBEAR) via Academy of Finland, decision number 334792. European Union's Horizon 2020 research and innovation programme under grant agreement No 689443 via project iCUPE (Integrative and Comprehensive Understanding on Polar Environments)" and funding from the 625 European Union's Horizon 2020 research and innovation programme under grant agreement No 821205 (Understanding and reducing the long-standing uncertainty in anthropogenic aerosol radiative forcing, FORCeS). University of Helsinki and Academy of Finland support to ACTRIS infrastructure and INAR RI Ecosystems are gratefully acknowledged (decision numbers 329274, 328616 and 304460). The staff of the SMEAR II station are thanked for the valuable help.

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
