# Peer review of "Determination of free amino acids, saccharides and selected microbes in biogenic atmospheric aerosols - seasonal variations, particle size distribution, chemical and microbial relations"

_Atmospheric Chemistry and Physics, 2020_

## Referee Comment (RC1) · Pierre Amato (Referee) · 1 Dec 2020

This work reports the size-resolved characterization of aerosol particles collected in a forested area in Finland (SMEAR II station). Samples were collected using cascade impactors and the different size fractions (<1 $\mu$m, 1-2.5 $\mu$m, 2.5-10 $\mu$m and > 10 $\mu$m) were analyzed independently for amino acids, saccharides, total DNA and total fungi, total bacteria and Pseudomonas, along with COVs measured online, with an attempt to disentangle the contribution of biological particles (PBAP) to the chemical content. In general, highest values are found in the 2.5-10 $\mu$m size. The data were combined with meteorological variables and analyzed through several statistical approaches (correla-

tion, regression, clustering), in order to reveal trends involving PBAP.

General comments:

- The dataset is interesting and original. This was analyzed exclusively through statistics, looking for trends between the different size fractions and between the different variables investigated, which is rather consistent with the underlying objective to identify a signature of specific PBAP. However, the absolute values themselves are neither discussed nor positioned respect to literature. It would be interesting to have a paragraph for discussing these, independently from trends.

- The title is probably too general and somehow inappropriate, as this is more about the interrelationships between variables than about the characterization itself. Also, is there any evidence that the particles looked for are indeed exclusively primary? and biological?

- Figure 1 and/or Tables S8-S11 could include a line with the sum of all fractions, which would thus correspond to the total aerosols load.

- I have a major concern with qPCR data and these are fundamental in this study. First, there is not even a mention of the genes targeted. Second and most importantly (I want to emphasize here that this is the reason why I recommended not going further with publication), the results: values around 1-10 genes (supposedly 16S and 18S rRNA)/m3 of air are reported, indicating the presence of 10 cells/m3 at the very most, which is absolutely not consistent when the literature reports orders of magnitude higher values around 105-106 copies/m3 in much more remote contexts (see notably (Dommergue et al., 2019; Šantl-Temkiv et al., 2017; Tignat-Perrier et al., 2019, 2020). Even the same authors reported incomparable values in previous publication (Helin et al., 2017), so either the data themselves are not valid as they largely underestimate the actual situation, or it could be that the unit used is wrong, or again that there was a mistake in the conversion to air volumes. It would be interesting to have indicated somewhere the cycle thresholds used for quantifications. - This work is basically a

repeat (improved?) of that published in 2017 by Helin et al, with different approaches and added with new variables like saccharides. There are at several occasions (auto)-plagiarism of this reference in the experimental section (maybe acceptable there (?)).

- The later reference is barely cited in the results and discussion section. However the present work would probably benefit to be positioned in context, with the findings discussed respect to previous ones.

- The choice of targeting in particular Pseudomonas among the humongous biodiversity that exists in the air must be justified. This is probably not obvious for everyone... Also, it might be useful to specify that Pseudomonas is a genus of bacteria at least in the introduction, this might not be obvious for every readers of ACP and it is presented as a distinct category.

- Unless I missed something, Table S1 and Figure 1 and Tables S8-S11 are the same data. However there are many inconsistencies, for instance the max values indicated for DNA, Pseudomonas and AA appear different from those in the figure. Can you check for any error and make the appropriate corrections.

- There is no mention of the results concerning control filters used for correcting chemical data: can you provide some information on what was found, if any contaminant was detected, and how the correction was done? Were there any such controls for microbiology (in addition of negative qPCR controls)?

Specific comments:

- L21 and throughout the manuscript: Specify which gene when mentioning gene copy numbers as it has no sense without this information.

- In Figure 1, the labels PM 2.5 and PM 10 are misleading as these are actually not PM 2.5 or PM 10 in the sense PM < 2.5 or < 10, but rather PM1-2.5 and PM2.5-10.

- L34: "...the influence of microbes...": The term "influence" suggests active intervention, is this what is meant? or does this rather refer to the contribution to the pool of

chemical compounds? This should be clarified by modifying "influence" if appropriate.

- L58: What is meant by "the role of amino acids in the atmosphere"? "impact" might be more appropriate?

- L77: "Viruses can be frequently found in the airborne..." state? (word missing)

- Section 2.3: the latin names of trees must be italicized.

- L 193-195: italicize latin organisms' names

- L233: Pearson correlations were used. Was the normality of data verified?

- Section 3.6: (link between microbiology and VOC): aerosols for microbiological analyses were collected at 23m, above the canopy, while VOCs were screened by PRT-MS inside the canopy at 8.4 m above ground. Why this discrepancy? And how could this had influenced the data? It is known that above-canopy and below-canopy air can be decoupled and can have different signatures (Gabey et al., 2010; Jocher et al., 2020).

References:

Dommergue, A., Amato, P., Tignat-Perrier, R., Magand, O., Thollot, A., Joly, M., Bouvier, L., Sellegri, K., Vogel, T., Sonke, J. E., Jaffrezo, J.-L., Andrade, M., Moreno, I., Labuschagne, C., Martin, L., Zhang, Q. and Larose, C.: Methods to investigate the global atmospheric microbiome, Front. Microbiol., 10, doi:10.3389/fmicb.2019.00243, 2019.

Gabey, A. M., Gallagher, M. W., Whitehead, J., Dorsey, J. R., Kaye, P. H. and Stanley, W. R.: Measurements and comparison of primary biological aerosol above and below a tropical forest canopy using a dual channel fluorescence spectrometer, Atmospheric Chemistry and Physics, 10(10), 4453–4466, doi:10.5194/acp-10-4453-2010, 2010.

Helin, A., Sietiö, O.-M., Heinonsalo, J., Bäck, J., Riekkola, M.-L. and Parshintsev, J.: Characterization of free amino acids, bacteria and fungi in size-segregated atmospheric aerosols in boreal forest: seasonal patterns, abundances

and size distributions, Atmospheric Chemistry and Physics, 17(21), 13089–13101, doi:https://doi.org/10.5194/acp-17-13089-2017, 2017.

Jocher, G., Fischer, M., Šigut, L., Pavelka, M., Sedlák, P. and Katul, G.: Assessing decoupling of above and below canopy air masses at a Norway spruce stand in complex terrain, Agricultural and Forest Meteorology, 294, 108149, doi:10.1016/j.agrformet.2020.108149, 2020.

Šantl-Temkiv, T., Amato, P., Gosewinkel, U., Thyrhaug, R., Charton, A., Chicot, B., Finster, K., Bratbak, G. and Löndahl, J.: High-Flow-Rate Impinger for the Study of Concentration, Viability, Metabolic Activity, and Ice-Nucleation Activity of Airborne Bacteria, Environ. Sci. Technol., 51(19), 11224–11234, doi:10.1021/acs.est.7b01480, 2017.

Tignat-Perrier, R., Dommergue, A., Thollot, A., Keuschnig, C., Magand, O., Vogel, T. M. and Larose, C.: Global airborne microbial communities controlled by surrounding landscapes and wind conditions, Sci Rep, 9(1), 1–11, doi:10.1038/s41598-019-51073-4, 2019.

Tignat-Perrier, R., Dommergue, A., Thollot, A., Magand, O., Amato, P., Joly, M., Sellegri, K., Vogel, T. M. and Larose, C.: Seasonal shift in airborne microbial communities, Science of The Total Environment, 716, 137129, doi:10.1016/j.scitotenv.2020.137129, 2020.

---

## Referee Comment (RC2) · Anonymous Referee #2 · 22 Dec 2020

The study aimed to identify correlations between the concentration of primary biological aerosol particles (especially bacteria, fungi and Pseudomonas qPCR-based concentrations) and different atmospheric compounds (free amino acids, saccharides, VOCs) using different statistical approaches. Other environmental data such as meteorological data were also analyzed. The authors collected airborne particles of four different particle size fractions in the Finland boreal forest using a PM10 sampler. The dataset is relatively large (84 samples) and interesting. The results revealed several correlations but the discussion remained elusive.

Main comments

[Figure]

-The authors used polycarbonate membranes to collect airborne particles of different sizes. Still, to characterize atmospheric chemistry, quartz fiber filters are mostly used in the literature due to their high retention rates (Innocente et al., 2017, Dommergue et al., 2019, Samake et al., 2019). Do the authors think that the use of these filters could have impacted the observed chemical concentrations?

-Could the authors specify why a relatively low volume of around 100 m3 has been used to collect airborne particles?  100 m3 is a low air volume considering the low microbial biomass in the air, especially if half of the filters has been used for molecular biology analyses.

-The authors have chosen to estimate the concentration of the Pseudomonas bacterial genus (based on qPCR gene copy number). Could the authors specify why they have chosen this bacterial genus? Is the choice based on some hypotheses or expectations that have not been specified in the text?

-The authors specified for each method what percentages of correct classification of the samples it provides, but what does a correct classification mean?  And to what extend is it useful (i.e. what does it mean for the non-classified samples) and used in the result interpretation?

-L93: why this whole methodological paragraph? These methods are widely used in atmospheric chemistry and microbial ecology.

-The results and discussion section showed little discussion and limited literature references. It is not clear if the authors expected specific correlations between airborne microbes and atmospheric chemical compounds. Like specified by the authors in the Introduction section, lots of chemical compounds are produced and emitted by microorganisms. Still, the sampling site is located within a forest in which microorganisms are really abundant (on trees, plants, in the soil. . .). These microorganisms (not necessarily airborne) would produce these chemical compounds that could become airborne, so why correlations between airborne microorganisms and chemical compounds would

be expected? Are the authors suggesting that atmospheric chemical compounds are emitted or composed airborne microorganisms?

In the same way, L530 ("In most of the cases, exception of methacrolein, these compounds were reported to be produced and emitted to the atmosphere by Pseudomonas (Effmert et al., 2012)."), do the authors suggest that Pseudomonas bacteria present in the air emit these compounds in the air? Pseudomonas bacteria present in the different ecosystems composing the forest (on the trees, plant leaves etc.) would also emit these compounds in the air.

Some recent papers relevant in the domain and that investigated the relationship between atmospheric chemistry and microorganisms are not referenced such as:

Samake, A., Jaffrezo, J.-L., Favez, O., Weber, S., Jacob, V., Canete, T., Albinet, A., Charron, A., Riffault, V., Perdrix, E., Waked, A., Golly, B., Salameh, D., Chevrier, F., Oliveira, D. M., Besombes, J.-L., Martins, J. M. F., Bonnaire, N., Conil, S., Guillaud, G., Mesbah, B., Rocq, B., Robic, P.-Y., Hulin, A., Le Meur, S., Descheemaecker, M., Chretien, E., Marchand, N. and Uzu, G. : Arabitol, mannitol, and glucose as tracers of primary biogenic organic aerosol : the influence of environmental factors on ambient air concentrations and spatial distribution over France, Atmos. Chem. Phys., 19(16), 11013–11030, doi:10.5194/acp-19-11013-2019, 2019.

Samake, A., Jaffrezo, J.-L., Favez, O., Weber, S., Jacob, V., Albinet, A., Riffault, V., Perdrix, E., Waked, A., Golly, B., Salameh, D., Chevrier, F., Oliveira, D. M., Bonnaire, N., Besombes, J.-L., Martins, J. M. F., Conil, S., Guillaud, G., Mesbah, B., Rocq, B., Robic, P.-Y., Hulin, A., Meur, S. L., Descheemaecker, M., Chretien, E., Marchand, N. and Uzu, G. : Polyols and glucose particulate species as tracers of primary biogenic organic aerosols at 28 French sites, Atmos. Chem. Phys., 19(5), 3357– 3374, doi:10.5194/acp-19-3357-2019, 2019.

Innocente, E., Squizzato, S., Visin, F., Facca, C., Rampazzo, G., Bertolini, V., Gandolfi, I.,Franzetti, A., Ambrosini, R., Bestetti, G., 2017.Influence of seasonality, air mass orig-

inand particulate matter chemical composition on airborne bacterial community structure in the Po Valley, Italy. Sci. Total Environ. 593–594, 677–687.

-L248: The sentence "Multiple linear regression, [. . .], was used to evaluate the effect of the microbial species on the chemical composition of the aerosol particles." How linear regression could evaluate this? A correlation does not mean a cause-effect relationship, and even if it was the case, could the atmospheric chemical composition affect airborne microbial species composition and not the other way around? The part 3.4 is called "Influence of the concentration of atmospheric gases, aerosol, meteorological and environmental parameters on the microbiological and chemical composition of the aerosol particles". Could it be the other way around?

-L419: Part 3.5 "Potential elucidation of chemical signals from microbes in aerosol particles", what does it mean? Could the authors try to make clearer subtitles in the Result and Discussion section.

Specific comments

-L19: In the abstract the authors wrote "The contribution of pollen, plant fragments, spores, bacteria, algae and viruses to PBAPs is well known." while it appears that the literature does not say so. The quantitative contribution of all these different PBAPs at any specific location is unknown.

-L47: The reference Reponen et al., 2001 ("Aerodynamic versus physical size of spores: Measurement and implication for respiratory deposition") is about the respiratory deposition of spores. Could the authors explain how it is related to the sentence on the potential long residence time of PBAPs?

-L74: Please correct the sentence, for example like that: "uncultivable or dead microorganisms, as well as fragments of plant. . ."

-L77: Please correct the sentence ("in the airborne")

-L86: Please correct the sentence: either add a comma after fragments or remove

"Fungi".

-L90: Sentence not clear. Do allergenic processes in humans induce considerable economic losses?

-L112: Please correct the sentence "microbial species (bacteria, fungi and Pseudomonas)" is incorrect as neither bacteria, fungi or Pseudomonas are species.

-L156 and L126: Please avoid repetitions in the Material and Methods section, for example regarding the different fraction sizes and the PM used.

-L339: Please could the authors explain what "growing mechanism" means.

-L341: Please explain "These results can be caused by the large number of potential PBAP sources present at the sampling site and their different contribution to the atmospheric aerosols." Please provide more details about what you think when giving suggestions (here regarding the potential other sources). The same shall apply for L80 ("under the influence of different environmental factors") and for L537 ("However, additional emission sources might not be discarded.") and the authors should add references.

-L463: "This could be explained by the role of the AA on the oxidant induced DNA damage of the organisms (Cantoni et al., 1992)." Do the authors suggest that the presence of histidine AA in the air would "kill" airborne microorganisms? Where would this AA come from?

-L472: "The presence of trehalose has been reported for a wide variety of microorganisms, including bacteria, yeast, fungi and insects (Elbein et al., 2003)." Do the authors mean "in"? Does trehalose a microbial cellular component?

-L486: "Pseudomonas had a clear correlation with Gly." Could the authors give in parentheses the R and pvalue associated, as a clear correlation is relatively subjective. In general, the authors should give the correlation values when talking about the presence of a correlation, even if the results are given in SI.

[Figure]

-L500: What do "accumulation processes" mean?

-L566: Please remove the comma at the end, add references and name the actual roles to inform the readers. "The role of plants in the production and emission of monoterpenes to the atmosphere is well known,."

-L568: Please make the sentence clearer "In this way, it is not possible to discard the contribution of bacteria to the monoterpene concentration or the role of environmental and meteorological parameters in the common emission of terpenes and bacteria to the atmosphere." Do "bacteria" mean airborne bacteria? Does "contribution" mean emission and/or composition?

References:

Innocente, E., Squizzato, S., Visin, F., Facca, C., Rampazzo, G., Bertolini, V., Gandolfi, I.,Franzetti, A., Ambrosini, R., Bestetti, G., 2017.Influence of seasonality, air mass originand particulate matter chemical composition on airborne bacterial community structure in the Po Valley, Italy. Sci. Total Environ. 593–594, 677–687.

Dommergue, A., Amato, P., Tignat-Perrier, R., Magand, O., Thollot, A., Joly, M., Bouvier, L., Sellegri, K., Vogel, T., Sonke, J.E., Jaffrezo, J.-L., Andrade, M., Moreno, I., Labuschagne, C., Martin, L., Zhang, Q., Larose, C., 2019. Methods to investigate the global atmospheric microbiome. Front. Microbiol. 10.

Samake, A., Jaffrezo, J.-L., Favez, O., Weber, S., Jacob, V., Canete, T., Albinet, A., Charron, A., Riffault, V., Perdrix, E., Waked, A., Golly, B., Salameh, D., Chevrier, F., Oliveira, D. M., Besombes, J.-L., Martins, J. M. F., Bonnaire, N., Conil, S., Guillaud, G., Mesbah, B., Rocq, B., Robic, P.-Y., Hulin, A., Le Meur, S., Descheemaecker, M., Chretien, E., Marchand, N. and Uzu, G. : Arabitol, mannitol, and glucose as tracers of primary biogenic organic aerosol : the influence of environmental factors on ambient air concentrations and spatial distribution over France, Atmos. Chem. Phys., 19(16), 11013–11030, doi:10.5194/acp-19-11013-2019, 2019.

Samake, A., Jaffrezo, J.-L., Favez, O., Weber, S., Jacob, V., Albinet, A., Riffault, V., Perdrix, E., Waked, A., Golly, B., Salameh, D., Chevrier, F., Oliveira, D. M., Bonnaire, N., Besombes, J.-L., Martins, J. M. F., Conil, S., Guillaud, G., Mesbah, B., Rocq, B., Robic, P.-Y., Hulin, A., Meur, S. L., Descheemaecker, M., Chretien, E., Marchand, N. and Uzu, G. : Polyols and glucose particulate species as tracers of primary biogenic organic aerosols at 28 French sites, Atmos. Chem. Phys., 19(5), 3357– 3374, doi:10.5194/acp-19-3357-2019, 2019.

---

## Author Response (AR1)

Reply to the comments on the manuscript acp-2020-1065:

Reviewer 1:

Comment: The dataset is interesting and original. This was analyzed through statistics, looking for trends between the different size fractions and between the different variables investigated, which is rather consistent with the underlying objective to identify a signature of specific PBAP. However, the absolute values themselves are neither discussed nor positioned respect to literature. It would be interesting to have a paragraph for discussing these, independently from trends.

Response: Absolute values obtained from the chemical and microbiological species were discussed at 3.2 section. In addition, these results were compared with those found in the literature. It was observed that concentration values were slightly low in comparison with those reported our previous study (Helin et al., 2017). However, it could be explained considering the sampling period used in this research, corresponding to autumn-winter with some episodes of snow. In addition, limitations diverted from the technique used for the determination of total DNA on the determination of low DNA concentrations were considered.

Comment: The title is probably too general and somehow inappropriate, as this is more about the interrelationships between variables than about the characterization itself. Also, is there any evidence that the particles looked for are indeed exclusively primary? And biological?

Response: We have followed the suggestion of the reviewer and modified the title of the manuscript. The new title is "Determination of free amino acids, saccharides and selected microbes in biogenic atmospheric aerosols - seasonal variations, particle size distribution, chemical and microbial relations".

Comment: Figure 1 and/or Tables S8-S11 could include a line with the sum of all fractions, which would thus correspond to the total aerosols load.

Response: Figure 1 and Tables S8-S11 were modified as requested.

Comment: I have a major concern with qPCR data, and these are fundamental in this study. First, there is not even a mention of the genes targeted.

Response: Thank you for this comment. We have now added information that the qPCR is targeting at bacterial 16S and fungal 18S ribosomal DNA.

Comment: Second, and most importantly, the results: values around 1-10 genes (supposedly 16S and 18S rRNA)/m3 of air are reported, indicating the presence of 10 cells/m3 at the very most, which is absolutely not consistent when the literature reports orders of magnitude higher values around 105-106 copies/m3 in much more remote contexts (see notably (Dommergue et al., 2019; Šantl-Temkiv et al., 2017; Tignat-Perrier et al., 2019, 2020). Even the same authors reported incomparable values in previous publication (Helin et al., 2017), so either the data themselves are not valid as they largely underestimate the actual situation, or it could be that the unit used is wrong, or again that there was a mistake in the conversion to air volumes. It would also be interesting to have indicated somewhere the cycle thresholds used for quantifications.

Response: Thank you for raising this concern. The units are lower than in our previous publication (Helin et al. Atmos. Chem. Phys., 17, 13089, 2017) and in other publications because the samples were collected in this study in September-November, and not during growing season. We also showed in our previous publication (Helin et al. Atmos. Chem. Phys., 17, 13089, 2017) that the PBAPs have seasonal variation and that in the winter samples collected in February and March the gene copy numbers were lower than in

samples collected during other months. In Helin et al. 2017 publication the extracted DNA amounts were also higher than in our current manuscript. In addition, we would like to point out that in the study described in Helin et al (2017) publication, the whole filter was used for the DNA extraction, but in the present study a half of the filter was used for the extraction of free amino acids and another half for the extraction of DNA. Due to low amounts of template, we could unfortunately not redo the qPCR with higher amount of sample. However, we could measure and determine the detection limit for DNA/filter.

In this work, the observed gene copy numbers varied between 0.1-200 16S gene copies/m3 in case of bacteria, 0.05-400 18S gene copies/m3 in case of fungi, and 0.05-6 16S gene copies/m3 in case of Pseudomonas. However, the obtained gene copy numbers varied highly depending on the filter size and sampling time, and thus the averaged values shown in Figures 2 and 3 seem low, since they represent average gene copy numbers across all the studied filter sizes in all the studied time points. Similar time-scale variation and size segregation were observed both in the present study and in the Helin et al. 2017 publication, but this was not emphasized here due to the different focus of the manuscript.

Comment: This work is basically a repeat (improved?) of that published in 2017 by Helin et al, with different approaches and added with new variables like saccharides. There are at several occasions (auto)-plagiarism of this reference in the experimental section (maybe acceptable there (?)).

Response: The methods used for the determination of chemical compounds, microbiological species and total DNA were based on those used in our previous research published in 2017 by Helin et al. with some modifications. In some cases, these modifications are substantial and needed to improve the analytical performance of the methodology or include new target compounds. In any case, experimental section was carefully reviewed and different sections containing potential auto-plagiarism were modified.

The later reference is barely cited in the results and discussion section. However the present work would probably benefit to be positioned in context, with the findings discussed respect to previous ones.

Response: Our previous research has been positioned in context in the introduction section. In our previous study amino acids, bacteria and fungi were determined in aerosol samples collected at SMEAR II station to stablish seasonal variations and size distributions. Additionally, the effect of few local meteorological factors and potential emission sources was also evaluated. Even though the observations of concentrations and distribution of different PBAPs are accumulating, there is still lack of a comprehensive understanding of the processes behind the different observations and on detailed chemical characterisation of the particles. In this new study, chemical compounds (amino acids and saccharides), microbial species (bacteria, fungi and Pseudomonas) and total DNA concentrations were determined and different statistical tools were used to clarify the relationship between particle size, environmental and meteorological conditions and the composition of biogenic aerosol particles. In addition, potential chemical fingerprints from microbes in biogenic aerosols. Finally, the potential connections between gas phase VOCs and the microbiological composition of the aerosol particles, bacterial, fungal or Pseudomonas gene copy numbers.

The choice of targeting in particular Pseudomonas among the humongous biodiversity that exists in the air must be justified. This is probably not obvious for everyone. . . Also, it might be useful to specify that Pseudomonas is a genus of bacteria at least in the introduction, this might not be obvious for every readers of ACP and it is presented as a distinct category.

Response: Introduction section has been modified according to reviewer's suggestions. It now includes the explanation for the selection of Pseudomonas.

Comment: Unless I missed something, Table S1 and Figure 1 and Tables S8-S11 are the same data. However there are many inconsistencies, for instance the max values indicated for DNA, Pseudomonas and AA appear different from those in the figure. Can you check for any error and make the appropriate corrections.

Response: The data in all Tables and in all Figures have been checked, revised and updated.

Comment: There is no mention of the results concerning control filters used for correcting chemical data: can you provide some information on what was found, if any contaminant was detected, and how the correction was done? Were there any such controls for microbiology (in addition of negative qPCR controls)?

Response: Thank you for this comment, we analyzed also control filters, in addition to the negative controls in qPCR. We have now added a description of the control filters to the method section: "In addition to the actual samples, we extracted DNA from eight blank filters, determined their DNA concentrations and used the extracts as templates in qPCR. All the blank filters were below detection limit in both DNA concentration assay with Qubit as well as in all of the three qPCR assays."

Specific comments:

Comment: L21 and throughout the manuscript: Specify which gene when mentioning gene copy numbers as it has no sense without this information.

Response: Specific genes used in the determination of the gene copy numbers have been included in the text.

Comment: In Figure 1, the labels PM 2.5 and PM 10 are misleading as these are actually not PM 2.5 or PM 10 in the sense PM < 2.5 or < 10, but rather PM1-2.5 and PM2.5-10.

Response: Figure 1 and tables have been modified and revised.

Comment: L34: "...the influence of microbes...": The term "influence" suggests active intervention, is this what is meant? or does this rather refer to the contribution to the pool of chemical compounds? This should be clarified by modifying "influence" if appropriate.

Response: The sentence has been clarified as requested.

Comment: L58: What is meant by "the role of amino acids in the atmosphere"? "impact" might be more appropriate?

Response: The sentence was modified according to the suggestions of the reviewer.

Comment: L77: "Viruses can be frequently found in the airborne..." state? (word missing)

Response: The sentence has been revised.

Comment: Section 2.3: the latin names of trees must be italicized.

Response: Latin names have been now written with italics.

Comment: L 193-195: italicize latin organisms' names.

Response: Latin organisms' names have been now written with italics.

Comment: L233: Pearson correlations were used. Was/how the normality of data verified?

Response: As stated in the text, standardized Skewness and Kurtosis tests were used for the evaluation of the normality of data distribution. Additional logarithmic transformation of the data was needed to ensure the normal distribution of the input variables. This point has now been stressed in the text.

Comment: Section 3.6: (link between microbiology and VOC): aerosols for microbiological analyses were collected at 23m, above the canopy, while VOCs were screened by PRT-MS inside the canopy at 8.4 m above ground. Why this discrepancy? And how could this had influenced the data? It is known that above-canopy and below-canopy air can be decoupled and can have different signatures (Gabey et al., 2010; Jocher et al., 2020).

Response: Sampling inlet for the impactor used for the collection of the samples was placed 5 m over the ground. Clarification was added to the section 2.3. 8.4 m above ground was the best option to collect the PRT-MS data of VOCs without any big differences.

Reviewer 2

Comment: The authors used polycarbonate membranes to collect airborne particles of different sizes. Still, to characterize atmospheric chemistry, quartz fiber filters are mostly used in the literature due to their high retention rates (Innocente et al., 2017, Dommergue et al., 2019, Samake et al., 2019). Do the authors think that the use of these filters could have impacted the observed chemical concentrations?

Response: Quartz fiber filters have many beneficial characteristics for airborne particles collection, such as chemical and biological stability, permeability, resistant to solvents, suitability for high temperatures, etc.). Although these filters adsorb very well fine particles., they adsorb also polar gas-phase compounds due to a high number of potential interactions (dipole-dipole, electrostatic, etc.) producing significant positive artifacts (Parshintsev et al., 2011). Polycarbonate and Teflon filters have been widely used for the determination of bacteria in air by optical techniques. They are also ideal for the collection of airborne particles without any microbiological contamination and it is possible to do background correction for chemical compounds concentrations by applying blank filters. In addition, the combination of polycarbonate and Teflon filters has been successfully used in our previous research (Parshintsev, J., Ruiz-Jimenez, J., Petäjä, T. et al. Comparison of quartz and Teflon filters for simultaneous collection of size-separated ultrafine aerosol particles and gas-phase zero samples. Anal Bioanal Chem 400, 3527–3535 (2011). https://doi.org/10.1007/s00216-011-5041-0) for the determination of microbes and amino acids in airborne samples.

Comment: Could the authors specify why a relatively low volume of around 100 m3 has been used to collect airborne particles? 100 m3 is a low air volume considering the low microbial biomass in the air, especially if half of the filters has been used for molecular biology analyses.

Response: It is true that the air sampling volume was quite low in comparison with other studies. However, sampling volume was minimized in this study to improve the sampling frequency resulted in enhanced number of data points available for the development of statistical models. In addition, the use of a half of filter for the determination of chemical compounds and another half for microbes had clear advantages in comparison with our previous research, allowing the determination of both chemical compounds and microbes from the same sample. Moreover, no problems were observed for most of the samples in terms of detection and/or quantitation limits, so most of compounds and microbes were successfully quantified except those in samples of smallest size particles.

Comment: The authors have chosen to estimate the concentration of the Pseudomonas bacterial genus (based on qPCR gene copy number). Could the authors specify why they have chosen this bacterial genus? Is the choice based on some hypotheses or expectations that have not been specified in the text?

Response: Introduction section was modified according to reviewer's suggestion to include the explanation for the selection of Pseudomonas in this research (see also answers to reviewer 1).

Comment: The authors specified for each method what percentages of correct classification of the samples it provides, but what does a correct classification mean? And to what extend is it useful (i.e. what does it mean for the non-classified samples) and used in the result interpretation?

Response: The approach developed in this manuscript is based on the development of a supervised pattern recognition models using the concentration of the chemical compounds and the gene copy number of the microbes as independent variables. The groups, established using a non-supervised pattern recognition approach were used as response factors and different carefully selected meteorological and environmental variables as input data. The performance (accuracy) of the supervised models was evaluated by cross validation using the training set. In addition, the validation set (samples not used for the development of the model) was used for the evaluation of the prediction capability of the models.

In this way, the percentage of correctly classified samples could be easily associated with a real differentiation of the samples into the selected groups based on the variables used for model development and their validity. This is especially important in the case of the validation set samples because they were not used in the model development. However, the presence of incorrectly classified samples, meaning samples not following the trend found for their group, could not be easily estimated using this approach. The incorrect classification could be related to experimental errors in the calculation of the chemical compounds and microbes, a missing data (problem detected in the case of PM<1 μm samples), or the potential influence of meteorological and environmental variables not considered in this study.

Section 3.4 was modified according to this discussion to clarify the meaning of the percentages achieved for the samples correctly classified.

Comment: L93: why this whole methodological paragraph? These methods are widely used in atmospheric chemistry and microbial ecology.

Response: Paragraph from L93 to L102 was removed from the introduction section according to reviewer's suggestion.

Comment: The results and discussion section showed little discussion and limited literature references. It is not clear if the authors expected specific correlations between airborne microbes and atmospheric chemical compounds. Like specified by the authors in the Introduction section, lots of chemical compounds are produced and emitted by microorganisms. Still, the sampling site is located within a forest in which microorganisms are really abundant (on trees, plants, in the soil. . .). These microorganisms (not necessarily airborne) would produce these chemical compounds that could become airborne, so why correlations between airborne microorganisms and chemical compounds would be expected? Are the authors suggesting that atmospheric chemical compounds are emitted or composed airborne microorganisms?

Response: It is well known that different chemicals, such as saccharides and amino acids are produced and in some cases emitted to the atmosphere by microbes via metabolic activities. Accordingly, it is extremely difficult to associate any specific correlation between the airborne microbe and the atmospheric chemical compounds. In our present study, we found correlations between chemical compounds and the gene copy numbers of the airborne microbes. The direct interpretation of the results might help with the elucidation of potential biomarkers for microbes in the biogenic aerosols. However, the relation between the airborne

microbes and their presence in the forest ecosystems should be also considered in the results. This point has been clarified in the first paragraph of 3.5 section.

Comment: In the same way, L530 ("In most of the cases, exception of methacrolein, these compounds were reported to be produced and emitted to the atmosphere by Pseudomonas (Effmert et al., 2012)."), do the authors suggest that Pseudomonas bacteria present in the air emit these compounds in the air? Pseudomonas bacteria present in the different ecosystems composing the forest (on the trees, plant leaves etc.) would also emit these compounds in the air.

Response: We agree with the reviewer that Pseudomonas bacteria present in the different ecosystems, composing of the forest (trees, plant leaves, etc.), can emit these compounds. In addition, bacteria in aerosol particles can emit different VOCs. This point has been clarified in the third paragraph of the 3.6 section.

Comment: Some recent papers relevant in the domain and that investigated the relationship between atmospheric chemistry and microorganisms are not referenced such as:

Samake, A., Jaffrezo, J.-L., Favez, O., Weber, S., Jacob, V., Canete, T., Albinet, A., Charron, A., Riffault, V., Perdrix, E., Waked, A., Golly, B., Salameh, D., Chevrier, F., Oliveira, D. M., Besombes, J.-L., Martins, J. M. F., Bonnaire, N., Conil, S., Guillaud, G., Mesbah, B., Rocq, B., Robic, P.-Y., Hulin, A., Le Meur, S., Descheemaecker, M., Chretien, E., Marchand, N. and Uzu, G. : Arabitol, mannitol, and glucose as tracers of primary biogenic organic aerosol : the influence of environmental factors on ambient air concentrations and spatial distribution over France, Atmos. Chem. Phys., 19(16), 11013–11030, doi:10.5194/acp-19-11013-2019, 2019.

Samake, A., Jaffrezo, J.-L., Favez, O., Weber, S., Jacob, V., Albinet, A., Riffault, V., Perdrix, E.,Waked, A., Golly, B., Salameh, D., Chevrier, F., Oliveira, D. M., Bonnaire, N., Besombes, J.-L., Martins, J. M. F., Conil, S., Guillaud, G., Mesbah, B., Rocq, B., Robic, P.-Y., Hulin, A., Meur, S. L., Descheemaecker, M., Chretien, E., Marchand, N. and Uzu, G. : Polyols and glucose particulate species as tracers of primary biogenic organic aerosols at 28 French sites, Atmos. Chem. Phys., 19(5), 3357– 3374, doi:10.5194/acp-19-3357-2019, 2019.

Innocente, E., Squizzato, S., Visin, F., Facca, C., Rampazzo, G., Bertolini, V., Gandolfi, I.,Franzetti, A., Ambrosini, R., Bestetti, G., 2017.Influence of seasonality, air mass origin and particulate matter chemical composition on airborne bacterial community structure in the Po Valley, Italy. Sci. Total Environ. 593–594, 677–687.

Response: References have been added to the text.

Comment: L248: The sentence "Multiple linear regression, [. . .], was used to evaluate the effect of the microbial species on the chemical composition of the aerosol particles." How linear regression could evaluate this? A correlation does not mean a cause-effect relationship, and even if it was the case, could the atmospheric chemical composition affect airborne microbial species composition and not the other way around?

Response: We agree with the reviewer that it is difficult, or even impossible, to evaluate the effect of the microbial species on the chemical composition of the biogenic aerosol particles using multiple linear regression. However, multiple linear regression analysis can be used to identify potential microbial biomarkers in airborne particles. This can be done by using the concentrations of chemical compounds as independent variables and the number of gene copies of the microbes as response variable (dependent variables). This kind of technique has some limitations because it is not possible to clarify if potential markers are components of the microbes, and are emitted from microbes or can be emitted from other sources such as plants, trees, animals, etc. However, it is clear that a combination of different variables (amino acids and saccharides) can be used to calculate the concentrations of the different microbes in the biogenic particles and therefore amino acids and saccharides can be used as potential microbial biomarkers.

Comment: The part 3.4 is called "Influence of the concentration of atmospheric gases, aerosol, meteorological and environmental parameters on the microbiological and chemical composition of the aerosol particles". Could it be the other way around?

Response: The main objective of this section is to evaluate the seasonal distribution of the chemical compounds (amino acids and saccharides) and microbes in the biogenic aerosol particles. It is clear that different variables can affect the emission of chemical compounds and microbes to the atmosphere and in this way to the formation and growing of aerosol particles. However, the potential influence of the particles (emitted or formed in the atmosphere) on these variables (at least some of them) should not be discarded. The title of the section has been modified to "Seasonal distribution of chemical compounds and microbes in biogenic aerosol particles".

Comment: L419: Part 3.5 "Potential elucidation of chemical signals from microbes in aerosol particles", what does it mean? Could the authors try to make clearer subtitles in the Result and Discussion section.

Response: The title of 3.4 and 3.5 sections were modified according to reviewer's suggestion and the discussion of the previous comments. Section 3.4 was modified to "Seasonal distribution of chemical compounds and microbes in biogenic aerosol particles"; and section 3.5 to "Identification of potential microbial biomarkers in biogenic aerosol particles".

Specific comments

Comment: L19: In the abstract the authors wrote "The contribution of pollen, plant fragments, spores, bacteria, algae and viruses to PBAPs is well known." while it appears that the literature does not say so. The quantitative contribution of all these different PBAPs at any specific location is unknown.

Response: The presence in PBAPs of pollen, plant fragments, spores, bacteria, algae and viruses is pretty well known and the sentence has been modified according to reviewer's suggestion.

Comment: L47: The reference Reponen et al., 2001 ("Aerodynamic versus physical size of spores: Measurement and implication for respiratory deposition") is about the respiratory deposition of spores. Could the authors explain how it is related to the sentence on the potential long residence time of PBAPs?

Response: We agree that this reference is not the most suitable to explain the PBAPs residence times in the atmosphere and the reference was replaced in the manuscript with more appropriate ones.

Comment: L74: Please correct the sentence, for example like that: "uncultivable or dead microorganisms, as well as fragments of plant. . ."

Response: The sentence has been modified in the manuscript.

Comment: L77: Please correct the sentence ("in the airborne")

Response: This sentence has been revised according to reviewer's suggestion.

Comment: L86: Please correct the sentence: either add a comma after fragments or remove "Fungi".

Response: This sentence has been corrected.

Comment:L90: Sentence not clear. Do allergenic processes in humans induce considerable economic losses?

Response: We agree that the effect of fungal induced allergic processes and diseases on the economy is difficult to quantify. Accordingly this sentence has been modified.

Comment: L112: Please correct the sentence "microbial species (bacteria, fungi and Pseudomonas)" is incorrect as neither bacteria, fungi or Pseudomonas are species.

Response: The term microbial species has been replaced with microbes in this sentence.

Comment: L156 and L126: Please avoid repetitions in the Material and Methods section, for example regarding the different fraction sizes and the PM used.

Response: 2.3 section has been modified to avoid unnecessary repetitions in the text.

Comment: L339: Please could the authors explain what "growing mechanism" means.

Response: This sentence has been modified to indicate that the evaluation of the potential correlations between different particle sizes, using as variables the individual chemicals and microbes, can be useful to identify the presence of a common emission source and/or the participation of these compounds and microbes in the growth of the biogenic aerosol particles.

Comment: L341: Please explain "These results can be caused by the large number of potential PBAP sources present at the sampling site and their different contribution to the atmospheric aerosols." Please provide more details about what you think when giving suggestions (here regarding the potential other sources). The same shall apply for L80 ("under the influence of different environmental factors") and for L537 ("However, additional emission sources might not be discarded.") and the authors should add references.

Response: We agree with reviewer and sentence L341 has now been removed from the text. In addition, the information that the different environmental factors can inactivate viruses in the atmosphere has been added to the text (L80). Finally, L537 was rephrased including a new reference.

Comment: L463: "This could be explained by the role of the AA on the oxidant induced DNA damage of the organisms)." Do the authors suggest that the presence of histidine AA in the air would "kill" airborne microorganisms? Where would this AA come from?

Response: The role of His as catalyzer to enhance DNA double-strand breakage in the presence of hydrogen peroxide has been reported in the literature (Cantoni et al., 1992). This could affect the results reported for the DNA concentration in the samples in the present study. However, the relation between the concentration of His and the potential dead airborne microorganisms is not clear. It should be emphasized as well that opposite to viruses, bacteria and fungi are able to auto-repair damages in their DNA.

Comment: L472: "The presence of trehalose has been reported for a wide variety of microorganisms, including bacteria, yeast, fungi and insects (Elbein et al., 2003)." Do the authors mean "in"? Does trehalose a microbial cellular component?

Response: Trehalose has been reported as a microbial cellular component in a wide variety of microorganisms, including bacteria, yeast, fungi and insects (Elbein et al., 2003). The sentence has been revised.

Comment: L486: "Pseudomonas had a clear correlation with Gly." Could the authors give in parentheses the R and pvalue associated, as a clear correlation is relatively subjective. In general, the authors should give the correlation values when talking about the presence of a correlation, even if the results are given in SI.

Response: R2 and p-values have been added in parentheses.

Comment: L500: What do "accumulation processes" mean?

Response: The term accumulation processes was replaced in the manuscript with the correct term nutrient accumulation process.

Comment: L566: Please remove the comma at the end, add references and name the actual roles to inform the readers. "The role of plants in the production and emission of monoterpenes to the atmosphere is well known,."

Response: The sentence has been clarified in the manuscript.

Comment: L568: Please make the sentence clearer "In this way, it is not possible to discard the contribution of bacteria to the monoterpene concentration or the role of environmental and meteorological parameters in the common emission of terpenes and bacteria to the atmosphere." Do "bacteria" mean airborne bacteria? Does "contribution" mean emission and/or composition?

Response: The sentence has been clarified according to reviewer's instructions. We want to say that it is not possible to discard common emission of terpenes and airborne bacteria under certain environmental and meteorological parameters. In addition, it is not possible to discard the contribution of monoterpene airborne bacteria emissions to the total monoterpene concentration in the air considering the very specific environmental and meteorological conditions present during the experiments.

---

## Referee Report (RR1)

Comments on Ruiz-Jimenez et al : « Chemical and microbiological characterization of primary biological aerosol particles at the boreal forest » revised into "Determination of free amino acids, saccharides and selected microbes in biogenic atmospheric aerosols - seasonal variations, particle size distribution, chemical and microbial relations"

I have to admit that I am a bit confused by the revision and concerned by the validity of the data and analyses. I do not follow the argument of the samples being collected in September-November in the present study and that would explain the low values of gene copy numbers. On the contrary the values should be much higher than those reported, referring to fig2 in Helin et al. (~1000 genes copies/m3 at least as expressed as bacteria/m3 in Helin). In addition, I do not really get the point regarding the fraction of filter used for explaining the low values, since the data were normalized to air volumes (was normalization to air volume somehow not linearly related to filter surface??).

It is indicated by the authors that "The data in all Tables and in all Figures have been checked, revised and updated". I am very puzzled here: all or almost all the values for amino acids, saccharides and gene copies have changed (increased, by different factors) in Figures 1, 2, Tables S8-S11 compared to the last version, whereas the data were not changed in other figures (Figure 3 and Figure S6 for example). The modifications done would need to be at least clearly listed and justified. Averages, standard deviations and CVs, min/max and ranges were modified (Tables S8-S11), but surprisingly neither skewness nor kurtosis were affected. I would then also expect the regression coefficients (based on parametric statistics) to change according to raw data modification, since the modifications of the data were not linear, but this is surprisingly not the case (Fig 4 and 5, S4, S7, etc).

For example: bacteria in PM2.5-10 (Table S10) were modified from 4.4 +/- 1.7 (average +/- SD) into 53.0 +/- 54.0 (factor of 12 between averages). Elsewhere (Table S8) Gln in PM2.5-10 was modified from 1.9 +/- 1.9 over into 5.6 +/- 4.7 (factor of ~3 between averages). However, the Pearson correlation between Bact and Gln is still the same as in the previous version (Figure S4).

References:

Helin, A., Sietiö, O.-M., Heinonsalo, J., Bäck, J., Riekkola, M.-L. and Parshintsev, J.: Characterization of free amino acids, bacteria and fungi in size-segregated atmospheric aerosols in boreal forest: seasonal patterns, abundances and size distributions, Atmospheric Chemistry and Physics, 17(21), 13089–13101, https://doi.org/10.5194/acp-17-13089-2017, 2017.

---

## Author Response (AR2)

**Manuscript acp-2020-1065 entitled "Determination of free amino acids, saccharides and selected microbes in biogenic atmospheric aerosols - seasonal variations, particle size distribution, chemical and microbial relations"**

**Editor Decision: Reject** (12 Feb 2021) by Aurélien Dommergue
Comments to the Author:
Dear Author,

I am sorry to inform you that after careful discussions with the reviewers I have decided to reject your manuscript.
The reviewer 2 raise important concerns about some results and more importantly underlined that both chemical data and gene copies data have been modified. This appears to question the validity of the study, and it is very suspicious that in the revised manuscript version, values in Tables and Figures were changed without any explanation. The statement "The data in all Tables and in all Figures have been checked, revised and updated " does not suffice to provide a thorough documentation why the values differ among the manuscript versions that will be all publicly accessible.

Given that the revision did not lead to a sufficiently improved manuscript and even more so, the data is doubtful, I have decided to reject this manuscript.

Sincerely yours
AD

Authors' response to Editor: Original data matrix, including actual concentrations of the chemical compounds and DNA and gene copies numbers of microbes, was transformed, using logarithmic transformation, to achieve normal data distribution as described in the text. This was compulsory for the adequate interpretation of the different statistical algorithms used in the manuscript. The data processing was successfully developed using the normal distributed data. This data (logarithmic transformed) was originally used for the development of Figures 1 and 2 and Tables S8-11. During original manuscript preparation for the clarification we decided to use the actual concentrations of the chemical compounds and the total DNA and the gene copy numbers of the microorganisms in these figures and tables, instead of the logarithmic transformed values. Then the actual concentrations and gene copy numbers without outliers were calculated from the normalized data. During this process an unidentified error, which affected the concentrations and the gene copy numbers, showed in figures and tables of the original manuscript, was encounted. This error was only detected thanks to Referee #1 during the reviewing process of Discussion manuscript version. We followed the valuable comments of Referee #1 and calculations based on the correct data plotted in Figure 1 were used for Figure 2 and Tables S8-11 of the revised MS version. We are very sorry, but unfortunately by our mistake, uncorrected Figures 3 and S6 were included in the revised version of the manuscript (corrected Figure versions in the end of this letter).However, all the calculations described in the manuscript were done with the logarithmic dataset taken from the first Discussion version, and the revision of values of Figures 1 and 2, and Tables S8-11 did not affect these calculations, nor discussions.

Answers to the comments of referees (Report #1 and Report #2)

**Report #1**
Submitted on 20 Jan 2021
**Referee #2:** Romie Tignat-Perrier, rom26.p@hotmail.fr

**Recommendation to the editor**

**1) Scientific significance**
Does the manuscript represent a substantial contribution to scientific progress within the scope of this journal (substantial new concepts, ideas, methods, or data)?

Outstanding Excellent **Good** Fair Low

**2) Scientific quality**
Are the scientific approach and applied methods valid? Are the results discussed in an appropriate and balanced way (consideration of related work, including appropriate references)?

Outstanding **Excellent** Good Fair Low

**3) Presentation quality**
Are the scientific results and conclusions presented in a clear, concise, and well structured way (number and quality of figures/tables, appropriate use of English language)?

Outstanding Excellent **Good** Fair Low

For final publication, the manuscript should be
**accepted as is**
accepted subject to **technical corrections**
accepted subject to **minor revisions**
reconsidered after **major revisions**
    I would be willing to review the revised paper, if the editor considers it necessary.
    I am **not** willing to review the revised paper.
**rejected**

**Suggestions for revision or reasons for rejection (will be published if the paper is accepted for final publication)**

**Report #2**
Submitted on 02 Feb 2021
**Referee #1**: Pierre Amato, pierre.amato@uca.fr

**Recommendation to the editor**

**1) Scientific significance**
Does the manuscript represent a substantial contribution to scientific progress within the scope of this journal (substantial new concepts, ideas, methods, or data)?

Outstanding Excellent Good **Fair** Low

**2) Scientific quality**
Are the scientific approach and applied methods valid? Are the results discussed in an appropriate and balanced way (consideration of related work, including appropriate references)?

Outstanding Excellent Good **Fair** Low

**3) Presentation quality**
Are the scientific results and conclusions presented in a clear, concise, and well structured way (number and quality of figures/tables, appropriate use of English language)?

Outstanding Excellent Good **Fair** Low

For final publication, the manuscript should be
**accepted as is**
accepted subject to **technical corrections**
accepted subject to **minor revisions**
reconsidered after **major revisions**
    I would be willing to review the revised paper, if the editor considers it necessary.
    I am **not** willing to review the revised paper.
**rejected**

**Suggestions for revision or reasons for rejection (will be published if the paper is accepted for final publication)**

Comments posted as supplement.

**Referee Report: acp-2020-1065-referee-report.pdf**

Comments on Ruiz-Jimenez et al: «Chemical and microbiological characterization of primary biological aerosol particles at the boreal forest»revised into "Determination of free amino acids, saccharides and selectedmicrobes in biogenic atmospheric aerosols -seasonal variations,particle size distribution, chemical and microbial relations"I have to admit that I am a bit confused by the revision and concerned by the validity of the data and analyses. I do not follow the argument of the samples beingcollected in September-November in the present study and that would explain the low values of gene copy numbers. On the contrary the values should be much higher than those reported, referring to fig2 in Helin et al. (~1000 genes copies/m3 at least as expressed as bacteria/m3 in Helin).In addition, I do not really get the point regarding the fraction of filter used for

explaining the low values, since the data were normalized to air volumes (was normalization to air volume somehow not linearly related to filter surface??).

References: Helin, A., Sietiö, O.-M., Heinonsalo, J., Bäck, J., Riekkola, M.-L. and Parshintsev, J.: Characterization of free amino acids, bacteria and fungi in size-segregated atmospheric aerosols in boreal forest: seasonal patterns, abundances and size distributions, Atmospheric Chemistry and Physics, 17(21), 13089–13101, https://doi.org/10.5194/acp-17-13089-2017, 2017.

Authors' response: The reason for the differences in gene copy numbers between 2014 and 2017 can be due to two aspects:
1)Climatic conditions/meteorological variables. As can be seen from Table below, the total number of particles in 2017 was around a half of that in 2014 supporting the difference between the results for genes copies/m3. Namely in Finland the annual variation in air microbial concentrations can be very high due to large differences in seasonal weather conditions.

| Year | Month | T air (degC) | Precipitation (mm) | UV a (W m-2) | UV b (W m-2) | T soil (degC) | GPP (μmol m-2 s-1) | TNP* |
|---|---|---|---|---|---|---|---|---|
| 2014 | Sep | 10.1 | 0.7 | 5.7 | 0.3 | 10.7 | 4.5 | 2632.1 |
| | Oct | 4.1 | 1.8 | 1.9 | 0.1 | 6.0 | 1.5 | 1515.1 |
| 2017 | Sep | 9.2 | 2.1 | 4.5 | 0.2 | 10.0 | 4.3 | 1334.9 |
| | Oct | 3.6 | 3.4 | 1.6 | 0.1 | 6.0 | 1.4 | 973.5 |

*TNP = total number of particles

2) Technical limitations regarding to the assay itself due to lower sample amount. We wanted to make chemical and DNA analysis from the same sample and that's why we divided the filter into two equal pieces, one for both analysis. This might give higher detection level as previously, but not affect the gene copy numbers/m3.

The data in all Tables and in all Figures have been checked, revised and updated". I am very puzzled here: all or almost all the values for amino acids, saccharides and gene copies have changed (increased, by different factors) in Figures 1, 2, Tables S8-S11 compared to the last version, whereas the data were not changed in other figures (Figure 3 and Figure S6 for example). The modifications done would need to be at least clearly listed and justified.

Authors' response: Original data matrix, including actual concentrations of the chemical compounds and DNA and gene copies numbers of microbes, was transformed, using logarithmic transformation, to achieve normal data distribution as described in the text. This was compulsory for the adequate interpretation of the different statistical algorithms used in the manuscript. The data processing was successfully developed using the normal distributed data. This data (logarithmic transformed) was originally used for the development of Figures 1 and 2 and Tables S8-11. During original manuscript preparation for the clarification we decided to use the actual concentrations of the chemical compounds and the total DNA and the gene copy numbers of the microorganisms in these figures and tables, instead of the logarithmic transformed values. Then the actual concentrations and gene copy numbers without outliers were calculated from the normalized data. During this process an unidentified error, which affected the concentrations and the gene copy numbers, showed in figures and tables of the original manuscript, was encounted. This error was only detected thanks to Referee #1 during the reviewing process of Discussion manuscript version.  We followed the valuable comments of Referee #1 and calculations based on the correct data plotted in Figure 1 were used for Figure 2 and Tables S8-11 of the revised MS version. We are very sorry, but unfortunately by our mistake, uncorrected Figures 3 and S6 were included in the revised version of the manuscript (corrected Figure versions in the end of this letter). However, all the calculations described in the manuscript were done with the logarithmic

dataset taken from the first Discussion version, and the revision of values of Figures 1 and 2, and Tables S8-11 did not affect these calculations, nor discussions.

Averages, standard deviationsand CVs, min/max and ranges were modified (Tables S8-S11), but surprisingly neither skewness nor kurtosis were affected.

Authors' response: Skewness and kurtosis test were developed using the logarithmic transformed data (this transformation was done to ensure the normal distribution of the data used in the statistical analysis), as described in the Table legends. No errors were found in the logarithmic data, therefore the values were not affected.

I would then also expect the regression coefficients (based on parametric statistics) to change according to raw data modification, since the modifications of the data were not linear, but this is surprisingly not the case (Fig 4 and 5,S4,S7,etc). For example: bacteria in PM2.5-10 (Table S10) were modified from 4.4 +/-1.7 (average +/-SD) into 53.0 +/-54.0(factor of 12 between averages). Elsewhere (Table S8) Gln in PM2.5-10 was modified from 1.9 +/-1.9over into 5.6 +/-4.7(factor of ~3 between averages). However, the Pearson correlation between Bact and Gln is still the same as in the previous version (Figure S4).

Authors' response: All the calculations described in the original and revised manuscripts were done with the logarithmic dataset taken from the first Discussion version, and the revision of values of Figures 1 and 2, and Tables S8-11 did not affect these calculations, nor discussions in the revised manuscript.

**Figure 3**

[Figure]

**Amino Acids**

[Figure]

**Saccharides**

**Microbes**

**DNA**

**Figure S6 (B)**

[Figure]